# A general computational design strategy for stabilizing viral class I fusion proteins

Karen J. Gonzalez [1], Jiachen Huang[2,3], Miria F. Criado[3,4], Avik Banerjee[2,3], Stephen M. Tompkins [2,3], Jarrod J. Mousa [2,3,5] & Eva-Maria Strauch [1,6,7] ✉

Many pathogenic viruses rely on class I fusion proteins to fuse their viral membrane with the host cell membrane. To drive the fusion process, class I fusion proteins undergo an irreversible conformational change from a metastable prefusion state to an energetically more stable postfusion state. Mounting evidence underscores that antibodies targeting the prefusion conformation are the most potent, making it a compelling vaccine candidate. Here, we establish a computational design protocol that stabilizes the prefusion state while destabilizing the postfusion conformation. With this protocol, we stabilize the fusion proteins of the RSV, hMPV, and SARS-CoV-2 viruses, testing fewer than a handful of designs. The solved structures of these designed proteins from all three viruses evidence the atomic accuracy of our approach. Furthermore, the humoral response of the redesigned RSV F protein compares to that of the recently approved vaccine in a mouse model. While the parallel design of two conformations allows the identification of energetically sub-optimal positions for one conformation, our protocol also reveals diverse molecular strategies for stabilization. Given the clinical significance of viruses using class I fusion proteins, our algorithm can substantially contribute to vaccine development by reducing the time and resources needed to optimize these immunogens.

Life-threatening viruses such as the human immunodeficiency virus (HIV)[1], Ebola virus[2], Pneumoviruses[3], and the pandemic influenza[4] and coronaviruses[5], use class I fusion proteins to induce the fusion of viral and cellular membranes and infect the host cell. During membrane fusion, class I fusion proteins refold from their metastable conformation (prefusion state) to the highly stable postfusion conformation to provide the energy mediating membrane fusion[6]. Their essential role in viral entry, as well as their location on the viral surface, makes class I fusion proteins one of the major targets of neutralizing antibodies and, thereby, an excellent candidate for vaccination[7]. However, while both

pre- and postfusion states are usually immunogenic, the labile prefusion state has been demonstrated to induce a more potent immune response in multiple viral families[8-12]. Consequently, the prefusion state has become an attractive vaccine candidate when its conformation can be maintained[8,13,14].

Based on structural analyses of the fusion mechanism, the stabilization of the prefusion conformation has been mainly achieved by preventing the release of the fusion peptide or by disrupting the formation of the coiled-coil structure characteristic of the postfusion state[15]. Two strategies have been particularly successful by either

[1]Institute of Bioinformatics, Franklin College of Arts and Sciences, University of Georgia, Athens, GA 30602, USA. [2]Department of Infectious Diseases, College of Veterinary Medicine, University of Georgia, Athens, GA 30602, USA. [3]Center for Vaccines and Immunology, College of Veterinary Medicine, University of Georgia, Athens, GA 30602, USA. [4]Department of Pathobiology, College of Veterinary Medicine, Auburn University, Auburn, AL 36849, USA. [5]Department of Biochemistry and Molecular Biology, Franklin College of Arts and Sciences, University of Georgia, Athens, GA 30602, USA. [6]Department of Pharmaceutical and Biomedical Sciences, College of Pharmacy, University of Georgia, Athens, GA 30602, USA. [7]Department of Medicine, School of Medicine, Washington University, St. Louis, MO 63110, USA. ✉e-mail: evas@wustl.edu

designing disulfide bonds at regions undergoing remarkable refolding or introducing proline substitutions to impair the formation of the central postfusion helices[13,16,17]. Other stabilization methods have been focused on identifying substitutions that increase favorable interactions or rigidify flexible areas in the prefusion structure. These methods either design cavity-filling substitutions[13,17,18], neutralize charge imbalances[13,17,18], or remove buried charged residues[19]. While the strategies mentioned so far have been effective, the lack of an automated approach has been a limitation, requiring extensive testing of variants. Notably, more than one hundred different protein variants were evaluated before finding a stable prefusion conformation of the Filovirus GP protein[20], the severe acute respiratory syndrome coronavirus 2 spike protein (SARS-CoV-2 S)[21], and the F protein from Hendra[22], Nipah[11], respiratory syncytial virus (RSV F)[17,18], human metapneumovirus (hMPV F)[23], and parainfluenza virus types 1–4[8].

To address these limitations, we developed a general computational approach where the protein's sequence is optimized for the conformation of interest (here, the prefusion state) while destabilizing the other conformation in silico. Our general strategy assumes that conformational rearrangements in class I fusion proteins can be frozen by introducing mutations that reduce the free energy of the prefusion form but do not benefit or better disrupt the postfusion state. While this negative design concept has been introduced before in multi-state design (MSD) protocols[24], our efforts to implement leading algorithms in class I fusion proteins, such as the MPI_MSD[25], evidenced poor sequence sampling. This is likely due to the extensive sequence-structure search space that must be evaluated when modeling both states simultaneously of these large, unpacked proteins. Therefore, we modified the design process by avoiding explicit negative design but using the undesired conformation as a guide to identify suboptimal positions. In a second combinatorial design step, we search for an optimal sequence for the conformation of interest within the subset of substitutions identified to improve the prefusion conformation while disfavoring the postfusion conformation. Using this two-step protocol, we can control the substitution rates by focusing on the most impactful changes according to computed energetic information. With this method, we successfully stabilized the prefusion state of several large proteins, namely the RSV F, the hMPV F, and the SARS-CoV-2 S proteins, illustrating its general use. Importantly, only 3–4 variants were necessary to evaluate experimentally, saving a significant amount of time and resources.

## Results

### Energy optimization of the prefusion over the postfusion conformation

As fusion proteins must accommodate multiple conformations to complete the membrane fusion process[6], they are not optimized for a singular state, and various energetically sub-optimal residues can be found within a given conformation. In our initial step, we identified sub-optimal positions for the prefusion conformation based on the protein's energetics or anticipated dynamics (Fig. 1). For the first approach, mainly used for the stabilization of the RSV F protein (based on the A2 strain, as published under the PDB 5W23[26]), we uncovered residue positions with contrasting stability between the pre- and postfusion conformations by calculating the energetic contribution of every residue to both states. Through in silico alanine mutagenesis, we quickly identified the contributions towards Gibbs free energy (ΔΔG) of a given residue's side chain, approximating the position's role in the stability of each conformation[27]. Negative ΔΔG scores (<−1.0 in Rosetta energy units, REU) indicated structural stabilization, while positive ΔΔG scores (>1.0 REU) suggested destabilization[27]. Consequently, by generating energetic maps for both states, we located residue positions exhibiting differential energetic contributions across conformations (Fig. 1 and Supplementary Data 1–3). In all our examples, about 40 − 50 positions displayed higher stability in the prefusion state than in the postfusion state (Supplementary Data 1–3).

For the hMPV F and SARS-CoV-2 S proteins, the protein dynamics was used as a secondary approach to identify sub-optimal positions. Here, we analyzed regions characterized by significant structural rearrangements between states, as indicated by motion levels of at least 10 Å. Both highly mobile residues and positions exhibiting contrasting energetics in response to alanine scanning were designated as designable. These designable positions were then exhaustively explored to find substitutions that could reverse the energy balance between states. As our main objective was to find mutations working synergistically rather than individually, all substitutions favoring the prefusion state over the postfusion conformation were subjected to a combinatorial design step (Fig. 1).

Unfortunately, while several sequences were found to lower the prefusion state while increasing the postfusion state energy, our initial combinatorial design step introduced a high number of mutations (~40 substitutions when focusing only on alanine-scanning identified positions and ~100 substitutions for the approach based on protein dynamics) (Supplementary Data 1–3). Consequently, to prevent changes in the immunological properties of the proteins, we aimed to decrease the number of designable positions. To achieve this, we compared the per-residue energetics of each introduced mutation with the respective native amino acids in both prefusion and postfusion states. Positions for which at least one mutation improved per-residue energies in the prefusion state by 0.5 REU were retained for further refinement, as well as positions displaying a notable destabilizing effect in the postfusion state (Supplementary Data 1–3).

About half of the initial designable positions were discarded by implementing the above filtering process (Supplementary Data 1–3). The remaining positions were used to reiterate the combinatorial design step, using only mutations identified during the first design phase. Following the combinatorial process, we reviewed all redesigned positions regarding their chemical interactions and the specific energy contributions reflecting these interactions. To prioritize the most stabilizing interactions, we selected up to ten positions where energy improvements were observed in hydrogen bonding for mutations with polar contacts, van der Waals interactions and total contact number for mutations optimizing molecular packing, and electrostatic interactions for mutations forming salt bridges. Additionally, we eliminated polar groups when buried (see Materials and Methods). Chosen mutations were remodeled in the pre- and postfusion structures to confirm their ability to reduce the prefusion state's energy while increasing the postfusion state's energy (Supplementary Fig. 1 and Supplementary Table 1).

### Biochemical characterization of RSV F, hMPV F, and SARS-CoV-2 S variants

After ranking all designed sequences based on their energy differences between states and identifying the most potentially stabilizing substitutions within each construct, we selected 3–4 variants for expression from each virus. For RSV F, one (R-1b) out of three designs was found to be a monodispersed, trimeric protein, as indicated by size exclusion chromatography (SEC) (Fig. 2a). The remaining two constructs aggregated in solution and presented negligible absorbance signals. Compared to the current prefusion-stabilized vaccine DS-Cav1, R-1b exhibited about a 3.5-fold increase in protein expression (Fig. 2a). For the hMPV F and SARS-CoV-2 S, two (M-104 and M-305) out of four and three (Spk-M, Spk-F, and Spk-R) out of three redesigned proteins behaved similarly, respectively (Fig. 2d, g). Notably, the spike designs enhanced the protein expression of their base construct (S-2P) by approximately 17-fold (Fig. 2g). This improved expression was on par with that of another prefusion-stabilized spike protein, the HexaPro construct[21] (Fig. 2g). Likewise, the hMPV F variant M-104 displayed an 8-fold increase in protein expression against its precursor (115-BV)[23], aligning the expression level of other highly stabilized prefusion protein, such as the DS-CavEs2 immunogen[28] (Fig. 2d).

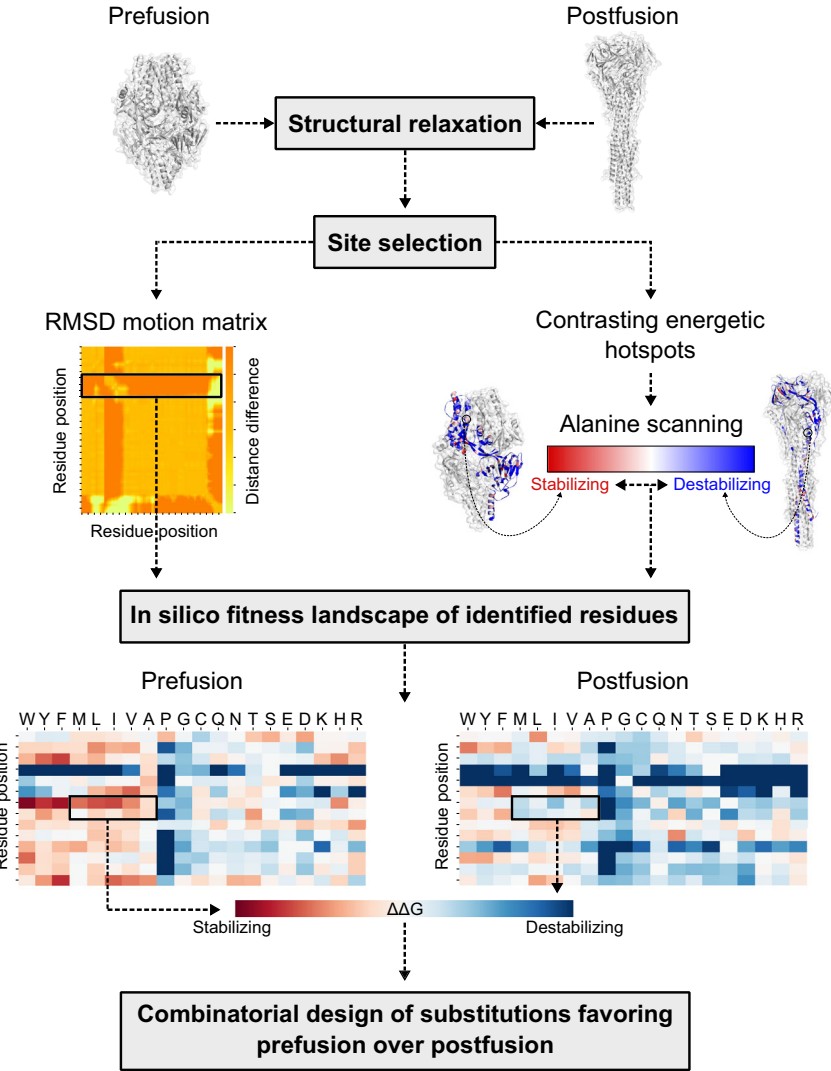

**Fig. 1 | Computational design overview.** Step-by-step diagram illustrating the key components of redesigning viral class I fusion proteins. Both pre- and postfusion conformations are input into the pipeline to identify substitutions that favor the prefusion state over the postfusion conformation. RMSD Root-mean square deviation, ΔΔG change in delta Gibbs free energy.

The conformational state of our purified designs was then evaluated based on the preservation of prefusion-specific epitopes of well-characterized neutralizing antibodies. All designs presented a prefusion-like structure as they bound tightly to prefusion-specific binders. For RSV F, we used antibodies D25[29,30] and AM14[30,31] (Fig. 2b and Supplementary Fig. 2a), and for hMPV F, antibodies MPE8[32] or 465[33] (Fig. 2e and Supplementary Fig. 2b). For SARS-CoV-2 S, we measured binding to the angiotensin-converting enzyme 2 (ACE2)[34] (Fig. 2h).

All the expressed proteins showed enhanced thermal stability compared to their respective precursor constructs (Fig. 2c, f, i, and Supplementary Table 1). The spike variants Spk-M and Spk-F displayed the most significant increase in melting temperature, with ~15 °C improvement over their parent construct S-2P[35] (Fig. 2i). Unlike S-2P[35], the Spk-M design preserved the prefusion conformation even after one hour of heating at 55 °C, as evidenced by its continued ACE2 binding at this temperature (Supplementary Fig. 3a). This level of prefusion stability was comparable to the highly stable HexaPro construct (Fig. 2i, Supplementary Fig. 3b), which was achieved by introducing several proline substitutions and experimentally evaluating 100 different variants[21].

A similar scenario was observed in the hMPV F variant M-104, which increased the melting point of its base construct by approximately 6.5 °C (Fig. 2f and Supplementary Table 1). M-104's melting temperature (61.5 °C) was equivalent to hMPV F variants containing up to two non-native disulfides, such as the DS-CavEs protein[28] (62.7 °C, Fig. 2f). Furthermore, M-104 maintained its antigenic integrity even after heating at 55 °C, as seen in constructs with designed disulfides like DS-CavEs and DS-CavEs2[28] (Supplementary Fig. 3c–e). While it is worth mentioning that the introduced disulfide bonds in DS-CavEs and DS-CavEs2 allow these proteins to preserve the prefusion state at higher temperatures than M-104 (Supplementary Fig. 3e), our results underscore how the strategic placement of electrostatic interactions can lead to highly stable proteins.

In the case of RSV F, the improvement of the prefusion state stability cannot be quantified since the wild-type sequence[26] was used as the starting construct. The inherent instability of this sequence substantially impedes its production as a soluble prefusion-state protein[17]; all purified RSV F molecules are found primarily in its postfusion state[36]. Therefore, obtaining the R-1b variant with a melting temperature of 62 °C revealed an effective optimization of the sequence to maintain the prefusion conformation (Fig. 2c). This result is especially relevant since no disulfide bonds were introduced, and stabilization was achieved only by enhancing non-covalent interactions. Notably, design R-1b proved to be antigenically intact even after heating at 55 °C (Supplementary Fig. 3f).

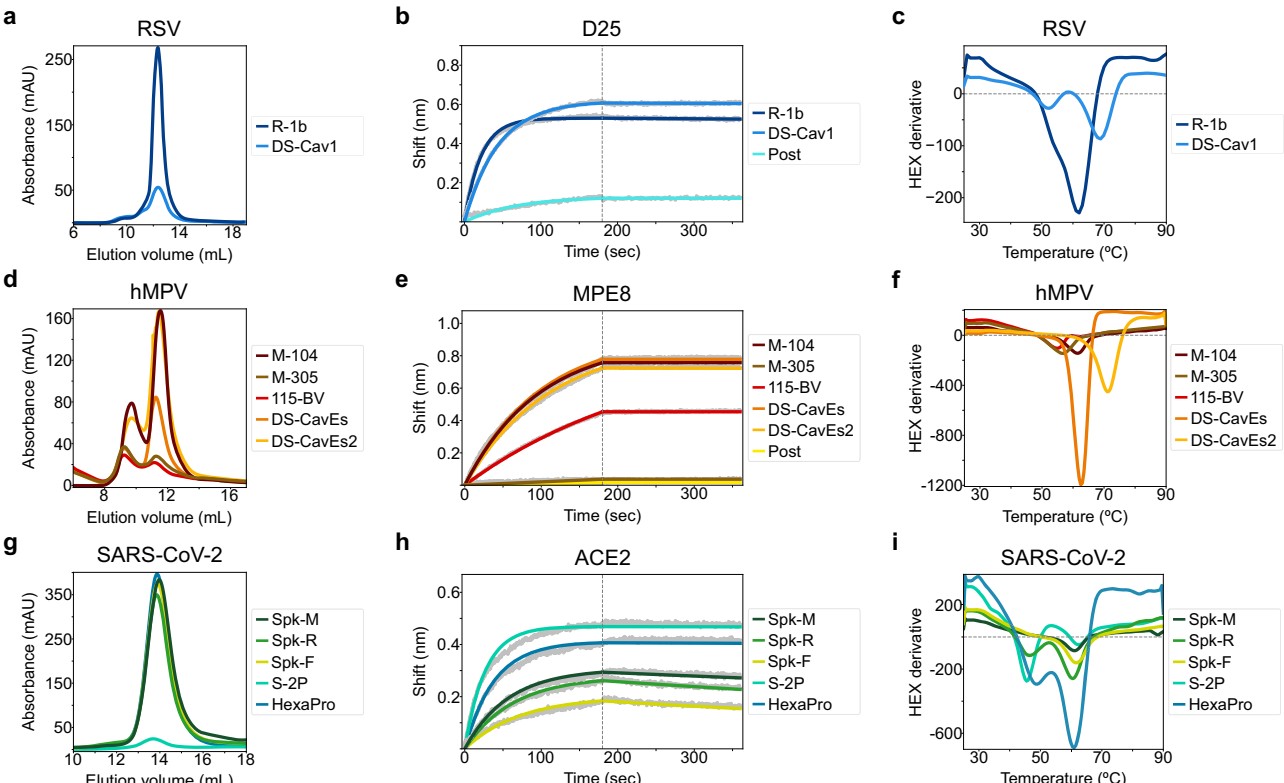

**Fig. 2 | Biochemical characterization of designed variants. a** Size-exclusion chromatography (SEC) of monodispersed RSV F variants. R-1b expression levels were compared to the RSV vaccine DS-Cav1. **b** Binding of design R-1b to the prefusion-specific antibody D25 compared to DS-Cav1 and the postfusion RSV A2 F (post). **c** Differential scanning fluorimetry (DSF) of design R-1b and DS-Cav1. DS-Cav1 was used to compare the stability of R-1b as the parental sequence of the latter is not prefusion-stabilized. **d** SEC of hMPV F variants. The expression levels of designs M-104 and M-305 were compared to their parent construct, 115-BV, and the next-generation immunogens DS-CavEs and DS-CavEs2[28]. **e** Binding of designed hMPV F variants to the prefusion-specific antibody MPE8 compared to 115-BV, DS-CavEs, DS-CavEs2, and the postfusion hMPV B2 F (post). **f** DSF of designed hMPV F variants, 115-BV, DS-CavEs, and DS-CavEs2. **g** SEC of SARS-CoV-2 S designs. Expression levels of designed proteins were compared to their parent construct, S-2P, and the next-generation immunogen HexaPro[21]. **h** Binding of designed SARS-CoV-2 S variants to ACE2 compared to S-2P and HexaPro. **i** DSF of designed SARS-CoV-2 S variants, S-2P, and HexaPro. Antibody binding assays show in grey the raw data, in colors the fitted curves, and in dotted lines the end of the association time. Binding constants are summarized in Supplementary Tables 2–4. Source data for all panels are provided as a Source Data file.

## Structure determination of leading RSV F, hMPV F, and SARS-CoV-2 S variants

Due to their enhanced expression levels and thermal stability, we selected the R-1b, M-104, and Spk-M designs for further characterization. Negative-stain electron microscopy (EM) confirmed that all three leading candidates presented a homogeneous trimeric prefusion morphology (Supplementary Fig. 4). This validation prompted us to investigate their atomic details using x-ray crystallography and cryo-EM. The crystal structure of the variants R-1b and M-104 verified their prefusion conformation at a resolution of 3.1 Å and 2.4 Å, respectively (Fig. 3a, b, and Supplementary Table 5). The accuracy of our computational predictions was reflected in the high structural similarity between the determined structures and the computational models, with root-mean-square deviations (RMSDs) of only 1.193 Å (405 Cα atoms) for R-1b and 0.53 Å (416 Cα atoms) for M-104. Furthermore, the 3D classification performed on the spike cryo-EM images also confirmed the prefusion integrity, with particles displaying one receptor binding domain (RBD) in the up conformation (Supplementary Fig. 5 and Supplementary Table 6). Solving the structure at a resolution of 3.7 Å (Supplementary Fig. 6) revealed that the S2 subunit, the only part engineered, closely matched the computational model, with an RMSD of only 1.345 Å (377 Cα atoms) (Fig. 3c).

While no significant perturbations were observed in all our variants overall, subtle differences at the antigenic site Ø were identified between R-1b and its parent RSV F protein. Specifically, the α4 helix in

R-1b displayed a bend toward residue D200 compared to the parent protein. Such variation has been noticed in prefusion structures, which appear to be intrinsically flexible when not bound to an antibody[37]; several prefusion-stabilized RSV F proteins, including the RSV vaccine DS-Cav1[13,17,18], happen to deviate in the same manner (Supplementary Fig. 7). In our design, we initiated from a bound RSV F structure (PDB: 5W23), stabilized in its prefusion conformation by co-crystallization with the antibody 5C4[26]. This apex flexibility suggests a potential avenue for future stabilization efforts, exploring whether a stabilized epitope would enhance the induction of antibodies binding to this site.

The crystal structures of R-1b and M-104 revealed that introduced substitutions followed their predicted stabilization mechanism by filling cavities or increasing intra- or interprotomer hydrogen bonds and salt bridges (Supplementary Fig. 8–9 and Supplementary Table 1). Especially precise agreement in rotamer orientation between our computational models and the experimental data was found in cavity-filling mutations, such as E60F in R-1b and the A159L and V203I in M-104 (Fig. 3a, b). Significant alignment in rotamer orientation was also observed in substitutions strengthening polar interactions, such as the N380K in R-1b and the L130D, V430Q, and V449D in M-104 (Fig. 3a, b, and Supplementary Fig. 9). Other mutations, such as the S150E and E487N in R-1b, did not interact with the predicted residues but still contributed to the prefusion stability by enhancing polar interactions at the protomers' interface (Supplementary Fig. 8). Lastly, we noted the alleviation of buried polar residues through their replacement with

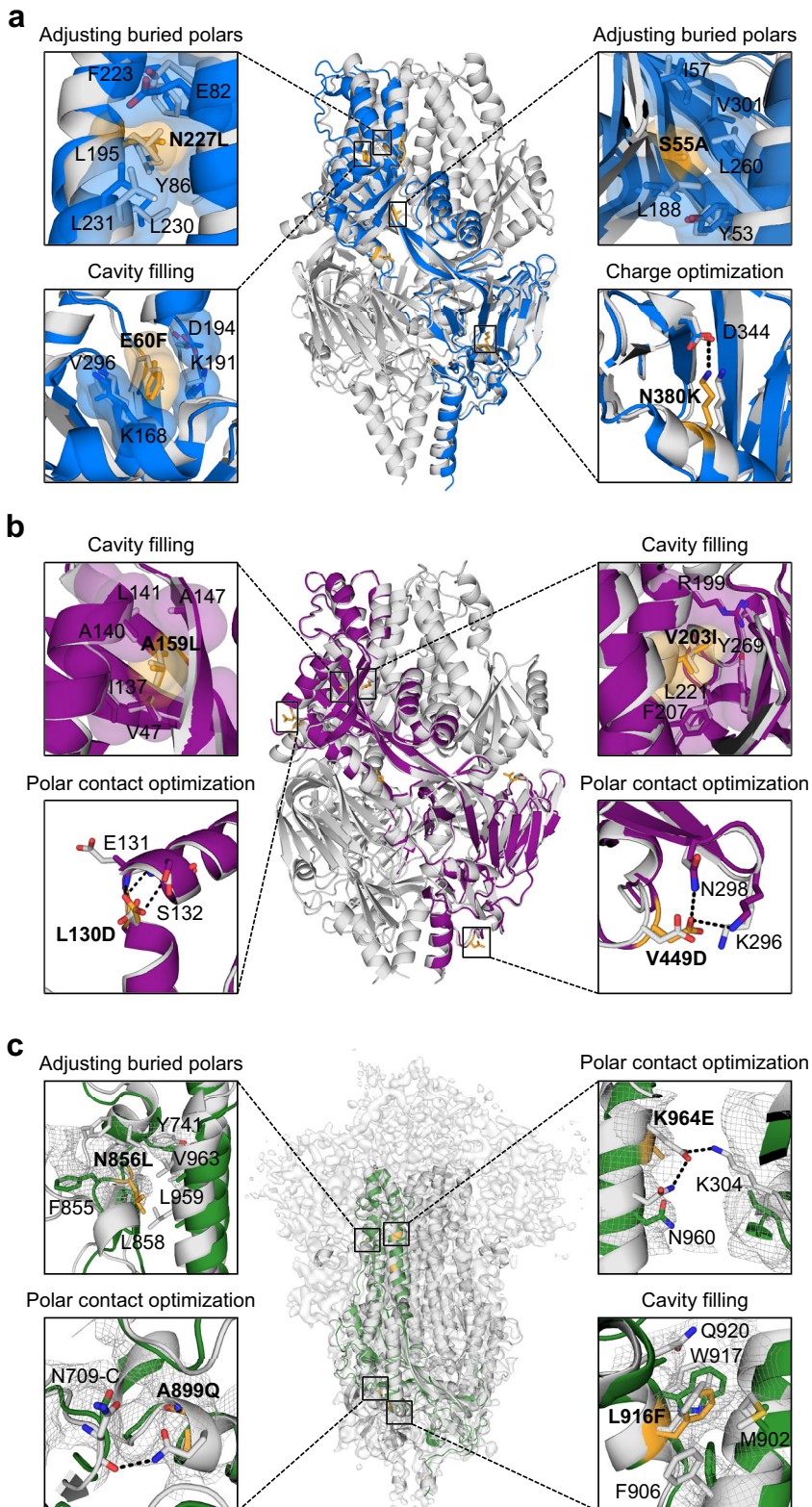

**Fig. 3 | Exemplary stabilizing substitutions of leading designs. a** R-1b. **b** M-104. **c** S2 subunit of Spk-M with cryo-EM map. The computational model of each protein is displayed as a trimeric structure in grey while the crystal structures and cryo-EM reconstruction model are displayed as monomeric structures in blue (RSV), purple (hMPV), or green (SARS-CoV-2). Each panel shows a magnified view of selected stabilizing substitutions, featured in yellow sticks, aligned with their computational model. Residues involved in packing changes are displayed with translucent molecular surfaces, and black dotted lines represent hydrogen bonds or salt bridges. As density is missing in the overall Spk-M map, we could not assign the precise location of the side chains. Therefore, we displayed existing density as a mesh representation to compare agreement with the computational model. The stabilization mechanism of all designed substitutions is presented in Supplementary Table 1.

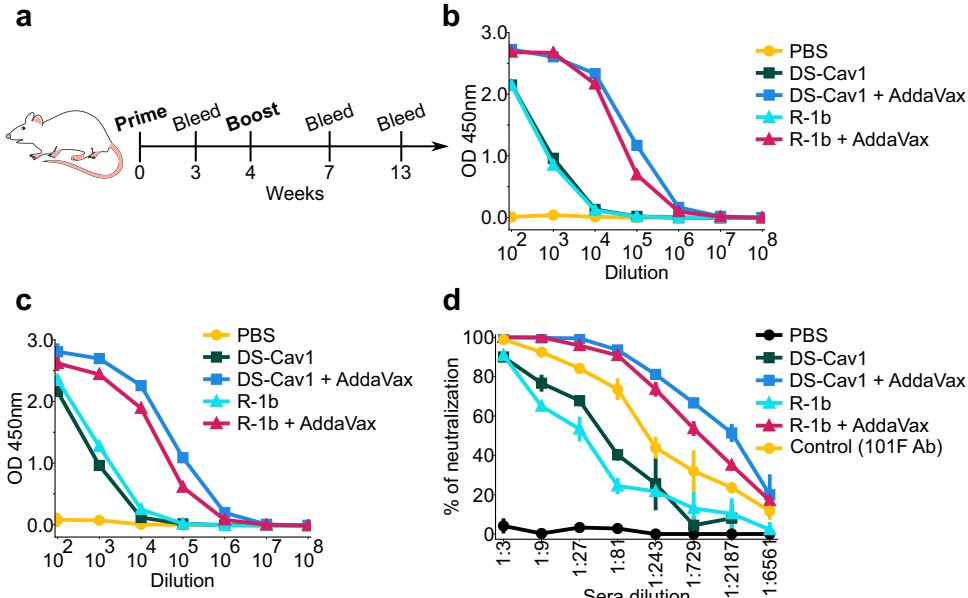

**Fig. 4 | Immunogenicity assessment of RSV F variants in a mouse model using 0.2 μg doses. a** Schematic diagram of vaccination study schedule. Mouse cartoon was created with ChemDraw 20.0[67] through licensing with the University of Georgia. **b** Serum RSV-specific IgG measured by ELISA three weeks post-boost. **c** Serum RSV-specific IgG measured by ELISA nine weeks post-boost. **d** Serum neutralization titers determined using RSV A (rA2 strain L19F) and sera from mice nine weeks post-boost. Ab stands for antibody. The markers on each line plot indicate mean values while the vertical lines represent the standard deviation. Values were calculated from three repetitions using pooled serum samples from mice within each immunization group (5 animals/group). Source data for panels **b**–**d** are provided as a Source Data file.

hydrophobic amino acids, improving the overall packing density. This effect was seen, for instance, after substituting the N227 with a leucine residue or removing the unsatisfied hydroxyl of S55 by replacing it with an alanine residue (Fig. 3a). Similar mutations were observed for the hMPV F protein. Finally, the Spk-M design achieved stabilization through four substitutions filling cavities and five substitutions increasing polar interactions at the S2 subunit, including three inter-protomer contacts (Fig. 3c, Supplementary Fig. 10, and Supplementary Table 1).

While most of the designed substitutions stabilized the prefusion state, the mutations N175R in R-1b, L130D in M-104, and T941D in Spk-M were additionally predicted to have a significant destabilizing effect in the postfusion conformation (Supplementary Fig. 11). In the post-fusion conformation, these residues are situated within the hallmark six-helix bundle, where charged and unsatisfied polar amino acids are highly unfavored and can potentially disrupt the core. As we had the postfusion-specific antibody 131-2A[38], we sought to test this hypothesis and confirm that we had not only stabilized the prefusion state but also destabilized the postfusion state. The diminished binding of the 131-2A antibody to R-1b after heating at 60 °C, a condition expected to convert the protein into its postfusion state, confirmed that we had achieved the design objectives we had set out to accomplish (Supplementary Fig. 3g).

### Immunogenicity of design R-1b

We selected the R-1b variant for a vaccination study due to the availability of a highly stable prefusion control, such as the DS-Cav1-based RSV vaccine[13]. Therefore, to investigate the effect of the introduced mutations on the RSV F immunogenicity, female BALB/c mice were vaccinated twice with either 0.2 or 2 μg of purified R-1b or DS-Cav1 with or without AddaVax adjuvant (Supplementary Table 7). Mice were bled at three and nine weeks post-second immunization (Fig. 4a). Sera analysis for binding to prefusion RSV F and RSV A2 neutralization revealed that R-1b induced similar levels of RSV F-specific antibody

titers (Fig. 4b, c, and Supplementary Fig. 12) and comparable neutralizing activity related to DS-Cav1 (Fig. 4d).

## Discussion

Detailed antibody response studies have illustrated that prefusion-stabilized class I fusion proteins are potent immunogens and promising vaccine candidates. This has been proven true for numerous viruses, including RSV[13,17,18], hMPV[28], parainfluenza[8], Nipah[11], MERS-CoV[16], and SARS-CoV-2[21]. Several of these immunogens have been developed by many iterative steps of manual structure-based design, with experimental evaluation often involving testing of hundreds of mutation combinations[8,11,17,18,21–23,39]. To alleviate this labor-intensive exploration, we automated one of the underlying principles behind their stabilization efforts, considering the biophysics of fusion proteins as the large irreversible switches they are. We have developed a computational approach that seeks to freeze the prefusion conformation by learning about sub-optimal contacts from its alternate conformation. Our algorithm systematically identifies these regions and their potential substitutions based on energy differences and relative motion between the two states.

Previous computational methods using multiple conformational states during protein design have encountered significant limitations due to their extensive computational requirements[25,40,41]. These demands stem from the concurrent sequence design and structural modeling of different protein states while introducing many mutations. The large size of full-length viral surface proteins further complicates continuous redesign and modeling, as increased computational expenses restrict efficient sequence sampling. Our computational strategy addresses these issues by focusing the sequence search exclusively on the desired conformation, while the undesired conformation delimits the specific substitutions to be sampled. Our approach reduces a multi-state design problem to a single-state design task, allowing a more comprehensive sampling of the sequence space.

Despite the computational advantages offered by our simplified design strategy, we acknowledge that the need for pre- and postfusion structures might present a limitation. Nonetheless, we believe that the remarkable advancements of recent structure prediction algorithms, such as AlphaFold[42] and RoseTTAFold[43], can facilitate the rapid modeling of both states. It remains to be evaluated whether the accuracy of these predictions is sufficient to guide our proposed design process. Furthermore, while our approach can easily discern energetic shifts resulting from several mutations, we have not explored its efficacy in the context of fewer or individual mutations. As various energy functions have shortcomings, we decided to use the combinatorial power only for a range of six to nine substitutions. Consequently, although our experiments yielded favorable outcomes within this range, it is likely that different numbers of mutations and combinations will also result in stabilized versions of the studied proteins.

Given the requirement for a solved prefusion structure, our hMPV F and SARS-CoV-2 S designs were based on pre-stabilized constructs. These mutations were essential to obtain their prefusion structures due to their inherent metastability. Their introduction was informed by successful strategies previously applied to RSV F[17] and the MERS-CoV Spike protein[16], where proline residues were intended to sterically hinder the expansion of the core three-helical bundle observed during the transition to the postfusion state. Thus, placing proline residues at the N-terminal end of this bundle effectively restricts the transition (as seen with P185 for hMPV and P986 and P987 for SARS-CoV-2 S). We opted to progress with these constructs rather than the wild-type sequence since, despite these mutations, the proteins remained notably unstable and were challenging to produce and store. We hypothesize that without these modifications, our protocol would likely have recognized the N-terminal position as a target for optimization due to the missing helix capping and potential disruption of the postfusion core. However, it is difficult to comment on this retrospectively as we do not have an unbiased starting structure for this purpose. Proline substitutions, either introducing or alleviating them, are challenging to model accurately since the extensive backbone modeling required can amplify modeling errors.

Another consideration in our designed proteins is the presence of surface-exposed substitutions. Given that alterations on the protein surface can potentially compromise the antigen's immunogenic properties, it is advantageous to minimize such mutations. In the cases of our R-1b and Spk-M designs, which introduced only two surface-exposed changes each (N175R and N380K in R-1b and T941D and P1143Q in Spk-M), this may not pose a problem, as confirmed by the RSV vaccination study. However, for hMPV, the M-104 construct incorporates four surface-exposed substitutions (A90N, L130D, V430Q, and V449D) that merit closer examination. Based on our estimation of per-residue energy changes (Supplementary Data 2), it is plausible that the mutation V430Q could be omitted from the M-104 design since its prefusion-stabilizing effect is only moderate compared to the other mutations (per-residue energy change of -2 REU, versus -5.8, -4.6 and -3.9 REU for A90N, V449D, and L130D, respectively). The remaining substitutions are likely significant contributors to stabilizing the prefusion structure, as they restrict the conformational flexibility of regions prone to refolding or reinforce the protein's quaternary structure by introducing salt bridges or hydrogen bonds.

In summary, the efficiency of our method was demonstrated across three different fusion proteins: RSV F, hMPV F, and SARS-CoV-2 S. In each case, only 3–4 variants were required to find a stable prefusion design. Furthermore, we validated the immunogenicity of one design in a mouse model, revealing similar in vitro neutralization and specific serum IgG patterns when compared to a current vaccine. As a result, our algorithm has the potential to impact the field of vaccine development by enabling rapid optimization of both class I fusion proteins and vaccine immunogens.

## Methods

All animal experiments were performed in accordance with the guidelines and approved protocols by the Institutional Animal Care and Use Committee at the University of Georgia, Athens, USA. The University of Georgia Animal Care and Use program is accredited by AAALAC International (Association for Assessment and Accreditation of Laboratory Animal Care), licensed by the USDA, and maintains an Assurance of Compliance with the Public Health Service.

All computational analyses were performed with the Rosetta version 2020.10.post.dev+12.master.c7b9c3e c7b9c3e4aeb1febab 211d63da2914b119622e69b.

### Structure preparation

The crystal structure of the RSV F protein in the prefusion (PDB: 5W23)[26] and postfusion (PDB: 3RRT)[36] conformations were refined with the Rosetta relax application using electron density data[44]. Density maps were generated from the corresponding map coefficients files associated with the PDB accession codes. These coefficients were transformed into density maps using the Phenix software version 1.15[45] and the option create map from map coefficients (region padding = 0, and grid resolution factor = 0.3333). To include the electron density in the refinement process, the density energy term was activated in the Rosetta scoring function with a weight of 20. This weight was selected given the low resolution of the density map and the starting structures. Four rounds of rotamer packing and minimization were performed during the relaxation protocol with gradual increases to the repulsive weight in the scoring function[46]. After five cycles of relaxation, the quality of the resulting models was evaluated with the Molprobity web service v4.5.1[47]. The structures with the lowest Rosetta energies and Molprobity scores were used for mutational analysis.

As described in the RSV F example, the hMPV F prefusion (PDB: 5WB0)[23] and postfusion (PDB: 5L1X)[48] conformations were relaxed using their respective electron density data. Due to the high resolution of the starting prefusion hMPV F structure, the weight of the density energy term was increased to 50 to encourage a good agreement with the density map. All other parameters and post-processing followed the RSV F example.

For the SARS-CoV-2 S protein, the input prefusion and postfusion structures and their corresponding cryo-electron microscopy (cryo-EM) maps were retrieved with the PDB accession codes 6VXX[49] and 6XRA[50], respectively. Since both structures were not completely solved, missing regions were modeled using the default comparative modeling protocol available in Rosetta[51]. However, the cryo-EM density maps of the input pre- and postfusion structures were integrated into the modeling process to avoid large deviations from the original configuration. Templates selected for prefusion modeling corresponded to the PDB IDs 6M0J[52] and the initial 6VXX, while the postfusion structure was modeled using 6LXT[53] and the initial 6XRA. Model selection was based on overall agreement with the starting structure and templates and a low Rosetta energy score. These homology models were then relaxed with the RosettaScripts[54] framework, incorporating fit-to-density parameters established for cryo-EM density[44]. Due to the high resolution of the starting structures, the refinement process was performed with a density scoring weight of 50 and three cycles of FastRelax[46] in cartesian space. The structures with the lowest Rosetta energies and Molprobity scores were used for mutational analysis.

### Selection of target positions to redesign

Residue positions to redesign were selected based on two independent approaches: a) contrasting energetic contributions to the pre- and postfusion conformations, and b) location on regions displaying drastic rearrangements between the pre- and postfusion states.

Amino acid positions with contrasting energetic contributions between conformations offer an opportunity to manipulate the

energetics of the conformational switch by allowing the optimization of one state while the other state is disfavored. Identification of these target spots was done by in silico alanine mutagenesis, where the energetic role of each residue on each conformation was estimated using the change in folding energy upon mutation. Consequently, residue positions displaying simultaneous stabilization of the prefusion state and destabilization of the postfusion state were selected as hotspots to redesign. Details about the selection process are described in the computational alanine scanning section.

As a secondary approach, all regions involved in the refolding process were chosen as targets to redesign. To identify these flexible areas, the root mean square deviation (RMSD) of each cα atom was calculated using their corresponding position in the pre- and postfusion structures. Both structures were structurally aligned prior to the analysis, and residue positions displaying motion levels of at least 10 Å were selected for redesign. Furthermore, residues flanking the highly mobile areas were also considered for redesign when their secondary structure differed between the pre- and postfusion states. Flanking residues were included until a set of 8 (SARS-CoV-2 S) or 16 (hMPV F) consecutive residues matched their secondary structure in the pre- and postfusion structures.

## Computational alanine scanning

A computational alanine scanning was performed on the pre- and postfusion states of RSV F, hMPV F, and SARS-CoV-2 S proteins to determine the energetic contributions of each amino acid to each conformation. Since the prefusion SARS-CoV-2 S contains domains absent in the postfusion conformation, alanine scanning in this protein was limited to shared regions between states. Using the Rosetta ΔΔG protocol cartesian_ddg[27], the backbone and sidechains around the position to be mutated were optimized in the cartesian space, and the change in folding energy (ΔG) was computed before and after each alanine substitution[27]. The contribution of every residue to stability was calculated in terms of ΔΔG scores (ΔG $_{mutant}$ - ΔG $_{wild\ type}$), where alanine changes holding ΔΔG scores < −1.0 were considered stabilizing substitutions while ΔΔG scores > 1.0 were considered destabilizing changes[27]. All calculations were repeated at least three times, and the average value among repetitions was used as the final ΔΔG score. To enhance the prefusion stability over the postfusion, regions presenting a stabilizing score in the prefusion conformation but not in the postfusion conformation were chosen to redesign. Specifically, designable positions were selected based on a) stabilizing ΔΔG score in the prefusion conformation and destabilizing ΔΔG score in the postfusion conformation, b) stabilizing ΔΔG score in the prefusion conformation and neutral ΔΔG score in the postfusion conformation, or c) destabilizing ΔΔG score in the postfusion conformation and neutral ΔΔG score in the prefusion conformation. To restrict the design process to the most relevant spots, positions meeting any of the above criteria were filtered based on an energetic difference of at least 0.7 (ΔΔG $_{postfusion}$ - ΔΔG $_{prefusion}$). Finally, since positions with native alanine are overlooked with this approach, all alanine-bearing spots were also included as targets for redesign.

## Computational protein design

To determine the amino acid identities likely to invert the energetics of the pre- and postfusion states, target positions to redesign underwent complete in silico saturation mutagenesis as described in the computational alanine scanning section. Subsequently, substitutions favoring the prefusion state over the postfusion state were chosen for combinatorial design through Rosetta modeling. To bias the design process towards mutations displaying a preference for the prefusion state, with a high energetic difference between states, the weight of each substitution was adjusted in the Rosetta energy function according to a fitness score. Our fitness score compiled the stabilization effect of one mutation in both pre- and postfusion states by subtracting the ΔΔG

prefusion score from the ΔΔG postfusion score (ΔΔG $_{postfusion}$ - ΔΔG $_{prefusion}$). Mutations favoring the prefusion state over the postfusion conformation were then characterized by positive fitness scores where higher values represented more significant energetic gaps between states. The fitness score was incorporated into the Rosetta score function through a residue-type constraint term derived from the FavorSequenceProfile mover. Since this mover was initially created to re-weight amino acid substitutions depending on their occurrence in a multiple sequence alignment, we have replaced the original position-specific substitution matrix (PSSM) input with a fitness score matrix. To follow the PSSM format, negative fitness scores were replaced by zero, and a 0.05 pseudo count was used for the log-odds scores calculation. After tuning the profile weights of each residue, allowed mutations at every amino acid position were defined as those with a fitness score greater than or equal to 0.7. For challenging targets such as the hMPV F and SARS-CoV-2 S, the threshold difference was increased to 2 to focus on the most significant substitutions. Likewise, beneficial mutations for both states were allowed only if the stabilization effect in the prefusion state was at least four units greater than in the postfusion state. Finally, the combinatorial sequence design was carried out by the FastDesign algorithm[55,56], enabling backbone and rotamer sampling. Upon conclusion, further optimization of a specific target spot was optionally done by applying FastDesign (all amino acids allowed) on residues neighboring 6 Å around the point of interest and limiting packing and minimization to a 12 Å sphere. The design process was initially performed on the prefusion conformation, and the resulting sequences were modeled on the postfusion structure for energetic comparisons.

## Selection of top designs

Promising designs were first sorted based on their Rosetta total energy score. Before comparison, the parent pre and postfusion structures were relaxed and energetically minimized using the same protocol as the designed models. Top candidates corresponded to designs showing a lower energy score in the prefusion state compared to the parent pre- and postfusion conformations. Analogously, the designed sequence in the postfusion state had to display a higher energy score than the parent postfusion conformation (Supplementary Data 1–3).

Considering that the initial round of combinatorial design introduced a substantial number of mutations, we aimed to reduce this number by pinpointing the most relevant positions for redesign based on their energy and types of interactions. To accomplish this, we measured all energy terms at a per-residue level, comparing the native residue with its mutated counterpart. Negative changes indicated that the mutation reduced the energy at the residue position relative to the native residue, while positive changes denoted an increase in energy compared to the native residue. However, recognizing that neighboring mutations may influence per-residue energies, we evaluated the average per-residue energy difference across all designs rather than individual construct values. This approach allowed us to identify promising target positions regardless of specific combinations of amino acids (Supplementary Data 1–3).

Designable positions for further analysis were selected based on the following criteria:

1. At least one mutation yielded a favorable energy improvement of 0.5 units or more compared to prefusion native residue (energy change in prefusion ≤ −0.5).
2. In instances where no mutations exhibited energy reduction in the prefusion state, positions were included if a significant destabilizing effect was evident in the postfusion state while the energy change in the prefusion state remained within 1 unit (energy change in postfusion ≥ 2.5 and energy change in prefusion <1).

3. Positions surface-exposed in both states were discarded unless a marked stabilizing effect of at least 2.5 units was observed in the prefusion state (energy change in prefusion ≤ −2.5).

Residue positions that fulfilled the above criteria were reintegrated into the Rosetta design algorithm for a second round of design. During this phase, only mutations identified in the initial design cycle were considered. Notably, both mutations with positive and negative energy changes were permitted, as the averaged energy calculation did not reflect the individual stabilizing effects of the mutations but rather their synergistic impact when combined with other changes.

Sequences resulting from the second round of design were retained based on their ability to improve the energy of the prefusion state while simultaneously elevating the energy of the postfusion state. These sequences were subsequently ranked by considering both their prefusion state energy and the energetic difference between the pre- and postfusion states. Designs within the top 50 candidates from both rankings were selected for per-residue energy examinations (Supplementary Data 1–3). Unlike the filtering process applied after the first iteration of design, the per-residue energy evaluation in the second selection process was performed for each design. Within each design, the number of introduced mutations was reduced by excluding all mutations displaying an energy change greater than (+) 0.5 units in the prefusion state (energy change in prefusion > 0.5). Exceptions to this criterion were only made when a solid destabilizing effect in the postfusion state was observed, while the energy change in the prefusion state remained under 1 unit (energy change in postfusion > 5 and energy change in prefusion <1).

At this stage, to avoid unintentional omission of potentially stabilizing individual substitutions due to the dominance of synergistic effects, we reviewed all the redesigned positions based on the following criteria:

1. Mutations presented in clusters, typically found in buried areas, were accepted if the cluster demonstrated tight, compact packing, accompanied by an increase in van der Waals contacts compared to the wild-type structure. Additionally, we ensured that the surrounding environment within a 10 Angstrom radius remained unperturbed regarding packing and hydrogen bonds.
2. In the case of the RSV and hMPV examples, we excluded hydrophobic substitutions, whether singular or grouped, at the protein interface. However, for the SARS-CoV-2 spike protein, characterized by its helical bundle arrangement in the inner interface, such mutations were only accepted if they contributed to improved packing and did not interfere with neighboring hydrogen bond networks.
3. For polar substitutions, we prioritized those featuring hydrogen bonds, confirmed through the hydrogen bond energy term in Rosetta. We placed particular emphasis on interactions occurring at the protomer interfaces, potential helix capping effects, and the preservation of a balanced charge distribution.
4. We refrained from accepting substitutions that retained the same charge as the original residue, e.g., substituting Arginine with Lysine, when considered as isolated mutations. However, if the mutation was part of a group, we evaluated them based on their potential to contribute to hydrogen bonding interactions.
5. We also considered secondary structure propensity.

The final selected mutations were modeled in the pre- and postfusion states to confirm their ability to reduce the prefusion state's energy while increasing the postfusion state's energy.

## Protein expression

The top 3–4 RSV F, hMPV F, and SARS-CoV-2 S computational designs, as well as the starting constructs hMPV F 115-BV[23] and SARS-CoV-2 S-2P[35], and the control variants RSV F DS-Cav1[13], postfusion RSV A2 F[36], hMPV F DS-CavEs and DS-CavEs2[28], postfusion hMPV B2 F[57], and the SARS-CoV-2 HexaPro[21] were expressed by transient transfection of FreeStyle 293-F cells (Thermo Fisher, Cat. # R79007) with polyethylenimine (PEI) (Polysciences). All computationally designed variants were produced in pCAGGS plasmids encoding the sequence of interest, a C-terminal T4 fibritin trimerization motif (Foldon), and a His6-tag. The designed RSV F constructs contained residues 1–105 and 137–513, and a flexible linker replacing the furin cleavage sites and p27 peptide (QARGSGSGR)[17]. Likewise, the designed hMPV F sequences included residues 1–95 and 103–472, a modified cleavage site ENPRRRR, and the A185P mutation[23]. Finally, the designed SARS-CoV-2 S variants followed the semi-stabilized SARS-CoV-2 S-2P protein sequence[35], with two proline substitutions at residues 986 and 987, and a GSAS linker replacing the furin cleavage site. All DNA sequences were codon optimized for human expression using the online tool GenSmart Codon Optimization[58]. Cells were cultured at 37 °C and 8% $CO_2$, and the culture supernatant was harvested on the third day after transfection. Proteins were purified by nickel affinity chromatography followed by size-exclusion chromatography (SEC) in phosphate-buffered saline (PBS) buffer pH 7.4. RSV F and hMPV F variants were SEC purified using a Superdex200 column (Cytiva), while SARS-CoV-2 S was purified with a Superdex6 column (Cytiva).

The angiotensin-converting enzyme 2 (ACE2) was expressed as an Fc-fusion[34] after transient transfection. The protein was purified using a Protein A agarose gravity column (Millipore Sigma) followed by SEC using an S200 column.

## Antigenic characterization

Bio-layer interferometry (BLI) was used to evaluate the structural and antigenic conservation of prefusion-specific epitopes. The prefusion-specific binders used for this purpose were the antibodies D25[29,30] (Cambridge Biologics, Cat. #01-07-0120) and AM14[30,31] (Cambridge Biologics, Cat. #01-07-0119) for RSV F, MPE8[32] and 465[33] (provided by Jarrod J. Mousa)[59] for hMPV, and the angiotensin-converting enzyme 2 (ACE2)[34] for SARS-CoV-2 S. All binders were immobilized on Protein A sensors (GatorBio) at a concentration of 15 nM (RSV F, and hMPV F), or 40 nM (SARS-CoV-2 S). Binding against expressed designs was tested at eight different protein concentrations starting from 200 nM (RSV F and hMPV F) or 400 nM (SARS-CoV-2 S) and decreasing by 1:2 dilutions. All solutions had a final volume of 200 μL/well using PBS buffer supplemented with 0.02% tween-20 and 0.1 mg/mL bovine serum albumin (BLI buffer). Biosensor tips were equilibrated for 20 min in BLI buffer before binder loading. Loading was then carried out for 180 s, followed by a baseline correction of 120 s. Subsequently, association and dissociation between the binders and designed variants were allowed for 180 s each. To validate the BLI results, binding with previous prefusion-stabilized proteins such as the RSV F DS-Cav1[13], hMPV F 115-BV[23], and SARS-CoV-2 S-2P[35] was used as positive controls, and binding with the postfusion constructs RSV A2 F[36], and hMPV B2 F[57] was used as negative controls. All assays were performed using a GatorPrime BLI instrument (GatorBio) at a temperature of 30 °C and frequency of 10 Hz. Data analysis was completed with the GatorOne software 1.7.28, using a global association model 1:1.

## Thermal stability

The thermal stability of the expressed variants was assessed by differential scanning fluorimetry (DSF). The samples were prepared by creating a solution containing 1.2 μL SYPRO orange fluorescent dye (Thermo Fisher) with 3 μL of 100 mM $MgCl_2$, 3 μL of 1 M KCl, and 3 μL of 1 M Tris (pH 7.4). The final solution volume was 60 μL with a protein concentration of 3.5 μM. A negative sample with no protein was also prepared as background control. All measurements were performed by triplicates using 20 μL of sample. The data was collected with a qPCR instrument (CFX Connect, BioRad) and a temperature

ramp from 25 to 90 °C with 0.5 °C increments. The melting temperature was determined based on the lowest point of the negative first derivative of the SYPRO Orange signal.

Antigenic preservation in variants displaying the highest melting temperature was further evaluated after one-hour incubation at 55 and 60 °C. This process was carried out in a thermocycler with a heated lid (T100, BioRad). The conservation of the antigenic sites was determined by binding to prefusion-specific binders, as described in the antigenic characterization section. Conversion to the postfusion state was also evaluated for the RSV F variant R-1b using the postfusion-specific antibody 131-2A[38] (Millipore Sigma, Cat. #MAB8599) and the postfusion RSV A2 F[36] as positive control.

## Negative-stain electron microscopy

Purified R-1b, M-104, and Spk-M (buffer-exchanged into 50 mM Tris pH 7.5 and 100 mM NaCl) were applied on carbon-coated copper grids (400 mesh, Electron Microscopy Sciences) using 5 μL of protein solution (10 μg/mL) for 3 min. The grid was washed in water twice and then stained with 0.75% uranyl formate (R-1b) or Nano-W (Nanoprobes) (M-104, and Spk-M) for 1 min. Negative-stain electron micrographs were acquired using a JEOL JEM1011 transmission electron microscope with a high-contrast 2K-by-2K AMT mid-mount digital camera.

## X-ray crystallization

The trimeric R-1b and M-104 proteins were concentrated to 14 mg/mL and 13.9 mg/mL, respectively, and crystallization trials were prepared on a TTP LabTech Mosquito Robot in sitting-drop MRC-2 plates (Hampton Research) using several commercially available crystallization screens. R-1b crystals were obtained in the Index HT (Hampton Research) in condition H6 (0.2 M Sodium formate, 20% w/v PEG 3,350), while M-104 crystals were obtained in the Crystal screen (Hampton research) in condition C10 (0.1 M Sodium acetate trihydrate pH 4.6, 2.0 M Sodium formate). Crystals were harvested and cryo-protected with 30% glycerol in the mother liquor before being flash-frozen in liquid nitrogen. X-ray diffraction data were collected at the Advanced Photon Source SER-CAT beamLine 21-ID-D (1 Å wavelength) at 77 Kelvin. Data were indexed and scaled using XDS[60]. A molecular replacement solution was obtained in Phaser[45] using the prefusion RSV F SC-TM structure (PDB: 5C6B)[17] or the prefusion hMPV F 115-BV (PDB: 5WB0)[23]. The crystal structures were completed manually in COOT 0.9.8.1[61], followed by subsequent manual rebuilding and refinement rounds in Phenix 1.15[45]. The data collection and refinement statistics are shown in Supplementary Table 5. The crystal structure of R1-b was refined to 3.1 Å (Ramachandran favored/allowed/outliers: 92.3%/7.5%/0.2%), while M-104 was refined to 2.4 Å (Ramachandran favored/allowed/outliers: 97.3%/2.7%/0%). It is important to highlight that the crystal structure of the R-1b protein displayed a relatively high R-free value, which proved challenging to decrease. We suspect that this increase may have been due to the presence of crystal twinning. Nevertheless, the R-free value obtained remains within an acceptable range, especially considering the resolution of 3.1 Å.

## Cryo-electron microscopy

Spk-M cryo-EM density data was obtained by the Eyring Materials Center at Arizona State University (ASU). Purified protein was diluted to a concentration of 0.35 mg/mL in Tris-buffered saline (TBS) and applied to plasma-cleaned CF-300 2/1 grids before being blotted for 3 s in a Vitrobot Mark IV (Thermo Fisher) and plunged frozen into liquid ethane. 3257 micrographs were collected from a single grid using an FEI Titan Krios (Thermo Fisher) equipped with a K2 summit direct electron detector (Gatan, Pleasantville, CA.). Data collection was automated with SerialEM with a defocus range of -0.8 to -2.6 μm in counting mode on the camera with a 0.2-second frame rate over 8 s and a total dose of 58.24 electrons per angstrom squared. Images were

processed using cryoSPARC V3.3.2[62] (Supplementary Fig. 5). Micrographs were patch motion corrected. After particle extractions, the blob picker was used, and 1,551,079 picking was manually adjusted to reduce blobs to 1,394,889 particles. After 2D classifications, the first four classes were ab initio reconstructed and heterogeneously refined. The most populated map was refined with homogenous refinement in cryoSPARC, resulting in a 3.72 Å map. The map was further processed in DeepEM[63]. The final map was aligned with a previously published SARS-CoV-2 S with one RBD domain up (PDB: 6VYB)[49] using UCSF Chimera-1.15[64]. Mutations and coordinate fitting were done manually using COOT 0.9.8.1[61], and structure optimization was achieved by iterative refinement using Phenix real space refinement[45] and COOT. The model and map statistics are presented in Supplementary Table 6.

## Mouse immunization

Six-to-eight-week female BALB/c mice were purchased and housed in individually ventilated Tecniplast SealSafe Plus caging. Mice were housed with a 12-hour photoperiod (light from 7:00 - 19:00 and dark from 19:00 to 7:00) with temperature set at 70 °F–72 °F and humidity monitored and maintained at 30–70%. Food and water were provided ad libitum. Animal sex was not considered in this study as the use of female mice follows a standard practice in RSV studies, facilitating high-titer replication of RSV in the lungs[65,66]. After the acclimation period, five mice per group were intramuscularly (i.m.) inoculated with a total of 100 μL of either purified DS-Cav1 or R-1b protein, with or without AddaVax adjuvant (50% v/v). The vaccination doses corresponded to 2 μg and 0.2 μg of the soluble proteins, as indicated in the Supplementary Table 7. Additionally, control experiments were carried out immunizing with only PBS. The inoculations were administered at weeks 0 and 4, following a Prime and Boost Vaccination protocol (Fig. 4a). Bleeds were collected from tail vein pre- and post-immunization (3, 7, and 13 weeks), and sera were analyzed by ELISA and neutralization assay.

## Measurement of IgG response by ELISA

Medium binding 96 wells microplates (Greiner Bio-One) were coated with 50 μL per well of DS-Cav1 or R-1b protein at 2 μg/mL at 4 °C overnight. Plates were washed in PBS/0.05% Tween 20 (Promega) and then blocked with blocking buffer solution (PBS/0.05% Tween 20 /3% non-fat milk (AmericanBio) /0.5% Bovine Serum Albumin (Sigma)) at room temperature for 2 h. Pooled serum from each group of mice pre- and post-different stages of immunization or control were inactivated at 56 °C per 1 h for subsequent serial dilution in blocking buffer. 100 μL per well of inactivated diluted sera were incubated in triplicate at room temperature for 2 h. Subsequently, three washes were performed, and plates were incubated with peroxidase-labeled goat anti-mouse IgG (1:3500) (SeraCare, Cat. #5220-0460) diluted in blocking buffer. After one hour of incubation at room temperature, plates were washed, and TMB substrate working Solution (Vector Laboratories) was added. After 10 min at room temperature, the reaction was stopped by adding 50 μL per well of Stop Solution for TMB ELISA (1 N H2SO4). Plates were then read on Cytation7 imaging Reader (BioTek) at 450 nm.

## RSV neutralization assay

Pooled serum samples from mice in each immunization group (5 animals/group) after vaccination and boost (13 weeks after the beginning of the experiment or prime vaccination/ 9 weeks after Prime and Boost vaccination) were diluted in Opti-MEM media (Thermofisher) in serial 3-fold dilutions. Antibody 101 F (provided by Jarrod J. Mousa)[59] was used as a positive control for virus neutralization starting at 20 μg/mL. Further, dilutions were mixed with 120 focus-forming units (FFU) of RSV A virus (strain: rA2 line19F) (kindly provided by Dr. Martin Moore) and incubated for 1 h at room temperature. Subsequently, RSV and sera/antibody dilutions were added to Vero E6 (ATCC #CRL-1586) monolayer (10^5 cells/well) in triplicate and incubated for one hour at

37 °C, gently rocking the plate every 15 min. Following the incubation, cell monolayers were covered with an overlay of 0.75% methylcellulose dissolved in Opti-MEM with 2% Fetal Bovine Serum (FBS) (Thermofisher) and incubated at 37 °C, 5% CO2. After four days, the overlay was removed, and wells were fixed with neutral buffered formalin 10% (Sigma) at room temperature for 30 min. Further, fixed monolayers were washed with water and dried at room temperature. An FFU assay was performed to identify the percentage of RSV neutralization. Briefly, wells were washed gently with PBS-0.05% Tween-20 (Promega) and incubated for one hour with anti-RSV polyclonal antibody (EMD Millipore Cat. #AB1128) diluted 1:500 in dilution buffer [5% Non-fat dry milk (AmericanBio) in PBS-0.05% Tween-20]. Plates were washed three times with PBS-0.05% Tween-20, followed by 30 min incubation of secondary antibody HRP conjugate rabbit anti-goat IgG (Millipore Sigma Cat. #AP106P) diluted 1:500 in dilution buffer. After incubation, wells were washed, and TMB Peroxidase substrate (Vector Laboratories) was added for 1 h at room temperature. The visualized foci per well were counted under an inverted microscope.

### Reporting summary

Further information on research design is available in the Nature Portfolio Reporting Summary linked to this article.

## Data availability

Atomic coordinates of the structures and cryo-EM map reported in this study were deposited in the Protein Data Bank under accession codes 7TN1 (R-1b), 8E15 (M-104), and 8FEZ (Spk-M), and in the Electron Microscopy Data Bank under accession code EMD-29035. Additional protein structures used in this study are available in the Protein Data Bank under accession codes 5W23, 3RRT, 5C6B, 5WB0, 5L1X, 6M0J, 6VYB, 6VXX, 6LXT, and 6XRA. The in silico energetic evaluations generated in this study are provided in Supplementary Data files. Source data are provided as a Source Data file. Source data are provided with this paper.

## Code availability

Scripts for generating designs are available on https://github.com/strauchlab/two-state-stabilization. Rosetta is available through licensing https://www.rosettacommons.org.

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

## Acknowledgements

We would like to thank Luki Goldschmidt for his support in maintaining the computing resources, including CyroSPARC. We would like to thank Dr. Dewight Williams at Arizona State University's Eyring Materials Center for taking the cryoEM data. This work was supported by federal funds from the National Institutes of Health grants R01AI140245 (KJG, EMS), National Institutes of Health grants R01AI143865 (JH, AB, JJM), National Institutes of Health grants Contract Numbers HHSN272201400004C, UGA-Emory Centers of Excellence for Influenza Research and Surveillance; CEIRS; (MFC, SMT), National Institutes of Health contract 75N93021C00018 (NIAID Centers of Excellence for Influenza Research and Response; CEIRR; (SMT). The Krios G2 was supported by the NFS MRI grant 1531991 to Dr. John Spence at Arizona State University's Eyring Materials Center.

## Author contributions

Conceptualization: KJG, EMS; Methodology: KJG, JH, MFC, AB, EMS; Formal analysis: KJG, JH, MFC, JJM, EMS; Writing–original draft: KJG, EMS; Writing–reviewing & editing: KJG, JH, MFC, SMT, JJM, EMS; Visualization: KJG, EMS; Supervision: SMT, JJM, EMS; Project administration: EMS; Funding acquisition: SMT, JJM, EMS.

## Competing interests

KJG and EMS are inventors of an ongoing US patent application No.18/404,463 (Recombinant viral class I fusion proteins and uses thereof. Patent status: Pending). The patent application has been filed by the University of Georgia covering stabilized viral class I fusion proteins, and methods of their use. The remaining authors declare no competing interests.
