## [Peer Review File · Nature Communications]

A general computational design strategy for stabilizing viral class I fusion proteinsREVIEWER COMMENTS

Reviewer #1 (Remarks to the Author):

This manuscript from Gonzalez et al. describes a multi-step, Rosetta-based design approach to stabilize the prefusion conformation of class I viral fusion proteins. This approach relies on the availability of accurate structural models of both the prefusion and postfusion conformations of a given protein and seeks to minimize the need for intensive experimental evaluation of putative stabilizing mutations. In an effort to validate this approach, the authors describe novel stabilizing mutations for the RSV F, MPV F and SARS-CoV-2 S fusogens. They go on to antigenically, biophysically and structurally validate that these proteins have been stabilized in their respective prefusion conformations. They also immunize mice with their prefusion-stabilized RSV F construct, R-1b, and show that it is able to elicit neutralizing titers comparable to a leading clinical candidate, DS-Cav1. The design strategy described in this manuscript will be of interest to vaccinologists, structural biologists and protein engineers. It also represents an initial effort towards partially automating the arduous and work-intensive process of prefusion stabilization and optimization. For these reasons, it is my opinion that this manuscript should be accepted for publication pending the revisions outlined below.

Major comments:

1. The manuscript would benefit greatly from an overall grammatical revision. Most of the data are compelling, but the numerous grammatical errors that can be found throughout the paper are a distraction from the ideas that are being presented.
2. Figure 2 should be revised to include the expression information that is currently shown in Fig S2 so that readers can see a direct comparison between the expression levels of the novel M-104/M-305 constructs alongside the 115-BV base construct. Furthermore, the equivalent expression information should be shown in Figure 2 for both the RSV-F and SARS-CoV-2 S base constructs and the next-generation immunogens which have been described for all three of these fusion proteins (DS-Cav1, DS-CavEs2 and HexaPro).

Minor comments:

1. The mutations A899Q and T941D which were incorporated into the Spk-M construct are similar to substitutions A899P and A942P which have been shown to enhance the prefusion stability of the SARS-CoV-2 spike as components of the HexaPro mutations. Can the authors speculate on the mechanism(s) by which their mutations are stabilizing S and whether there might be some similarities with these previously described HexaPro mutations?
2. Overall the structural studies appear to be sound from the PDB reports that have been provided, but the relatively high R-free value for the crystal structure of R-1b warrants some discussion or explanation

in the accompanying Methods section.

3. It would be worth explicitly mentioning that the RSV F base construct (PDB ID: 5W23) that was used for the design of R-1b was trapped in the prefusion conformation by the neutralizing antibody 5C4. The presence of this antibody is particularly relevant given the observation that the Domain 0 of their R-1b crystal structure displays some degree of conformational flexibility when unbound (see PMID: 31306469).

4. Similar to Major Comment #2, it would be interesting to see how previously described constructs such as DS-Cav1, DS-CavEs2 and HexaPro compare to these new designs in terms of their REU, as displayed in Fig S1.

5. Because the novel MPV F and SARS-CoV-2 S stabilizing mutations reported in this manuscript were layered on top of base constructs which had already undergone some degree of stabilization, some discussion of whether these new mutations would be sufficient to stabilize the prefusion conformations of MPV F and SARS-CoV-2 S in the absence of the 115-BV and S-2P mutations would be a welcome addition.

Reviewer #2 (Remarks to the Author):

Summary

In the study the authors have developed a computational method for designing stable viral pre-fusion constructs for a range of viruses and then validated in vitro and in vivo.

Major Points

1. More could potentially be done in the mouse model. It would be interesting to look at the impact on challenge with RSV infection – disease and peak viral load. It would also be interesting to compare against a less stable/ post fusion recombinant F protein to demonstrate that the approach is an improvement. Potentially compare to sera after infection too.

Minor Points

- Page 1 line 29-31 in abstract needs re-wording ‘allows identification and selection’.
- Page 12 line 5 says 105 cells/well. Presumably is supposed to be 10⁵
- Figure 4 and Mouse immunization section of methods (page 11, line 24)

Dosing schedule could do with being clearer. Presumably it is 2ug of the DS-Cav1 and 0.2ug of R-1b protein and these same doses at prime and boost. Figure legend on Fig 4 says ‘using 0.2ug doses. Which is it?’

Reviewer #3 (Remarks to the Author):

Gonzalez et. al. describe a general computational strategy for stabilizing class I fusion proteins by avoiding explicit negative design but using the undesired conformation as a guide to identify suboptimal positions and additional sequence optimization for the conformation of interest.

As the authors discuss in their conclusion, the approach needs both a prefusion and a postfusion structures, the approach is quick and generally applicable and brings an elegant solution to stabilizing class I fusion proteins.

A shortcoming of the paper is the lack of detail on the analysis.

For instance, in the first example that they describe (RSV F), the authors state that about 40 - 50 positions displayed higher stability in the prefusion state than in the postfusion state. It would be helpful if these data are shown in a Figure or supplementary figure or table. Only the 7 (best?) mutations are shown. The authors should try throughout the manuscript to show the complete analysis. Would it be possible to show a clear annotated larger part (or complete) heatmap? The data will allow the appreciation of the selection of the mutations and also allow comparison with known stabilizing point mutations that are previously described at e.g. positions 155, 190, 207, 215, 486 and 487.

The advantage of the approach is that very few variants were made and investigated, but a drawback is that the amount of mutations are relatively high and individual contribution of each mutation still needs to be studied which will increase the amount of variants to be generated. An immunogen preferentially contains a minimum amount of mutations. This problem should be addressed. Therefore, a systematic analysis of the contribution of each of the potentially stabilizing mutations still needs to be performed. Moreover, in the current study, large amount of potentially stabilizing mutations (7) are described but no individual comparison with other known stabilizing point mutations are shown. If a minimal amount of mutations need to be selected, the authors should rank the effect on expression and quality and compare that with known stabilizing mutations.

For their first example, RSV F, an energetic gap of at least 119 was selected. For RSV the variant of 119.1 REU was superior to the variants with higher REU. This means it is not straight forward to select an energetic gap and perhaps variants with lower gaps might have superior properties. Also for HMPV F, it was the variant with a lower REU gap that was superior over variants with highest REU gap. For RSV F R-1b was selected but no data is shown why this protein was superior over the others and how much better it was compared with the other RSV F candidates. Please add such data that justifies the selection. Apart from that, only four designs are shown. Could interesting designs have been missed based on the arbitrary energetic gap cut off? It would be helpful if a longer list of solutions would be listed in a supplementary table with lower energetic gaps. Perhaps there are solutions with fewer mutations that still result in a well behaving protein.

For HMPV, there are 7 differences between the best and second best. More analysis are needed to tease out the bare minimum of stabilizing mutations and to identify the mutations that really matter.

Page 3, line 41: HMPV F is compared to 115-BV. Perhaps a better comparison would have been the variant described in the Hsieh paper (Nat Comm 2022).

Page 3, last sentence: the current vaccines are not based on soluble S2P ectodomains but full-length variants. There is no evidence that Spk-M and F are superior to full-length proteins and whether they are better immunogens. Therefore, be more precise and delete the words 'current SARS-CoV-2 vaccine'

Page 4, line 36: The structural variation in the apex are large. Since the apex is the most important target

for neutralizing antibodies, more evaluation is needed to evaluate the impact. A detailed affinity measurement of D25 and other site 0 binders to R-1b and DS-Cav1 is important to evaluate the best design to stabilize this region in the most native state able to bind and induce apex antibodies.

The approach predicts mutations that destabilize the postfusion state. However, unlike e.g. VSV-G, the conformational change of the described proteins is unidirectional and none of these proteins are known to change back from postfusion to prefusion conformation. Therefore, although the calculations based on $\Delta\Delta G$ of prefusion and postfusion states is relevant and delivers successful stabilizing mutations, it is not because the postfusion structure is destabilized but because the prefusion structure is stabilized. Residues that the authors claim are postfusion disrupting are making favorable interactions in the prefusion conformation according to the structure data that the authors describe in their manuscript. Residues that are surface exposed in the prefusion conformation can apparently have a stabilizing effect on the prefusion conformation (although not preferred for vaccine components.) Authors should indicate which of the mutations are surface exposed and for immunogen design it would be helpful to show which of the exposed mutations can be omitted from the current designs.

Minor: Page 2, line 21-25: wrong reference to Cell Reports papers by Rutten et. al. The first reference (19) is likely Rutten et. al., Cell Reports 23, 584-595, 2018, in which mutations of buried charges is described for the first time.

A successful analysis described in the paper depends on the resolution of the structures that are used. Why was 5w23 used for the analysis for RSV prefusion F Is it the structure with highest resolution?

Reviewer #4 (Remarks to the Author):

Thank you for the opportunity to read and review the manuscript “A general computational design strategy for stabilizing viral class I fusion proteins.” The authors have developed a Rosetta-based computational approach that seeks to stabilize the prefusion conformation by learning about suboptimal contacts relative to its alternate postfusion conformation. The algorithm allows the identification of these regions, and their potential substitutions, based on energy differences and relative motion between the two states. The computationally designed sequences are then validated by studying the immunogenicity of one design in a mouse model and in vitro neutralization and specific serum IgG patterns compared to a clinical candidate for two other cases.

I am writing this review from the point of view of someone with expertise in computational protein design, engineering, and software development. I do not have the expertise regarding the experimental procedure included in the manuscript.

Thank you for sharing the code, which will ensure reproducibility and transparency. While I have thoroughly enjoyed reading the well-written manuscript, I have the following few comments and queries:

1. Page 8, line 27: Finally, the combinatorial sequence design was carried out by the FastDesign

Is the FastDesign with a fixed backbone, or flexibility in the backbone is allowed here?

2. Fig 2I: What do the two peaks of SPK-R in DSF plots signify?

3. Due to proximity to the cellular membrane, will there be any benefit of using membrane-informed energy functions or terms instead of energy functions for soluble proteins?

4. A few comments on the challenges and benefits of this strategy of protein vaccine design relative to other strategies might be helpful.

5. There are several papers where rosetta-based energy and structural measures are used to design new proteins. It is not very clear about the novelty of this approach. Mentioning the challenges faced by current methods and the improvements attained by this approach is needed.

We want to thank the reviewers for their thorough review of our manuscript. We included new experimental data to address their concerns and discussed several of their questions now within the main manuscript part.

Reviewer #1 (Remarks to the Author)

Summary

This manuscript from Gonzalez et al. describes a multi-step, Rosetta-based design approach to stabilize the prefusion conformation of class I viral fusion proteins. This approach relies on the availability of accurate structural models of both the prefusion and postfusion conformations of a given protein and seeks to minimize the need for intensive experimental evaluation of putative stabilizing mutations. In an effort to validate this approach, the authors describe novel stabilizing mutations for the RSV F, MPV F and SARS-CoV-2 S fusogens. They go on to antigenically, biophysically and structurally validate that these proteins have been stabilized in their respective prefusion conformations. They also immunize mice with their prefusion-stabilized RSV F construct, R-1b, and show that it is able to elicit neutralizing titers comparable to a leading clinical candidate, DS-Cav1. The design strategy described in this manuscript will be of interest to vaccinologists, structural biologists and protein engineers. It also represents an initial effort towards partially automating the arduous and work-intensive process of prefusion stabilization and optimization. For these reasons, it is my opinion that this manuscript should be accepted for publication pending the revisions outlined below.

Major comments:

1. The manuscript would benefit greatly from an overall grammatical revision. Most of the data are compelling, but the numerous grammatical errors that can be found throughout the paper are a distraction from the ideas that are being presented.

The manuscript has been reviewed and corrected for grammatical errors.

2. Figure 2 should be revised to include the expression information that is currently shown in Fig S2 so that readers can see a direct comparison between the expression levels of the novel M-104/M-305 constructs alongside the 115-BV base construct. Furthermore, the equivalent expression information should be shown in Figure 2 for both the RSV-F and SARS-CoV-2 S base constructs and the next-generation immunogens which have been described for all three of these fusion proteins (DS-Cav1, DS-CavEs2 and HexaPro).

We ordered the requested constructs and carried out their expression and purification. Simultaneously, we re-expressed and purified our constructs to ensure comparable conditions. The SEC profiles of the hMPV F and SARS-CoV-2 S base constructs, along with the next-generation immunogens DS-Cav1, DS-CavEs2, and HexaPro, have been presented in Figure 2. This figure also includes the DS-CavEs protein to facilitate a subsequent comparison of the thermal stability of M-104. Including DS-CavEs is particularly relevant because it has fewer non-native disulfides (two disulfides) than the DS-CavEs2 construct (four disulfides). This feature enables a more equitable melting point comparison with our M-104 design, as M-104 does not possess non-native disulfides.

It is worth noting that Figure 2 does not present the RSV F base construct. This omission is due to the protein's propensity to aggregate in solution, as its sequence was not optimized to retain the prefusion conformation. In consequence, the SEC profile of the RSV F base construct was not monodispersed, and the absorbance signal was negligible. This explanation was included on page 5, lines 8-9.

A detailed comparison of expression differences can be found on page 5, lines 9-18. Our leading constructs, M-104 and Spk-M, exhibited comparable expression levels to the next-generation immunogens DS-CavEs2 and HexaPro, respectively. On the other hand, the R1-b protein demonstrated a 3.5-fold increase in expression compared to DS-Cav1.

Minor comments:

1. The mutations A899Q and T941D which were incorporated into the Spk-M construct are similar to substitutions A899P and A942P which have been shown to enhance the prefusion stability of the SARS-CoV-2 spike as components of the HexaPro mutations. Can the authors speculate on the mechanism(s) by which their mutations are stabilizing S and whether there might be some similarities with these previously described HexaPro mutations?

Both residues 899 and 941 flank helical regions in the prefusion structure. These regions, although discontinuous in the prefusion state, they merge and form an extended helical structure within the core of the postfusion state. We hypothesize that mutations at these positions contribute to maintaining the prefusion state by creating a higher energetic barrier to the helical rearrangement.

In the prefusion structure, residue 899 is situated at the N-terminus of the α -helix comprised of residues 898-909. Our computational models indicated that the mutation A899Q could act as a helix-capping agent, important for preserving the helix's integrity. Additionally, A899Q can increase polar interactions at the protomer interface by hydrogen bonding with residue N709. These two interactions likely stabilize the prefusion helix and increase the energy needed to allow the helical rearrangement in the postfusion state. Similarly, the A899P in the HexaPro design plays an analogous helix-stabilizing role by capping the beginning of the helix. Furthermore, as proline residues are not favored on helical structures (Fujiwara, K., Toda, H. & Ikeguchi, M., BMC Struct Biol, 2012, doi: 10.1186/1472-6807-12-18), the formation of the postfusion helix is discouraged.

Similarly, position 941 in the prefusion state resides in a linker loop connecting two helices: the first formed by residues 913-940 and the second by residues 943-967. During the transition to the postfusion conformation, this linker region also becomes a helical structure. Like the A899Q substitution, our T941D substitution is predicted to stabilize the second helix by acting as a helix cap at its N-terminal end. Simultaneously, the mutation may restrict the flexibility of the linker loop and obstruct its rearrangement by hydrogen bonding with residue S943. Conversely, the A942P substitution in the HexaPro design appears to slightly disrupt the N-terminal conformation of this second helix (PDB 6xkl). Therefore, it is plausible that this substitution primarily stabilizes the prefusion structure by rigidifying the linker region and decreasing its propensity to form the postfusion helix.

2. Overall the structural studies appear to be sound from the PDB reports that have been provided, but the relatively high R-free value for the crystal structure of R-1b warrants some discussion or explanation in the accompanying Methods section.

We noticed that the dataset was twinned, which might have impaired the R-free. However, we believe that our R-free value is still within the acceptable range given the resolution of 3.1Å. We included this explanation on page 14, lines 40-43.

3. It would be worth explicitly mentioning that the RSV F base construct (PDB ID: 5W23) that was used for the design of R-1b was trapped in the prefusion conformation by the neutralizing antibody 5C4. The presence of this antibody is particularly relevant given the observation that the Domain 0 of their R-1b crystal structure displays some degree of conformational flexibility when unbound (see PMID: 31306469).

This information was included on page 6, lines 25-26.

4. Similar to Major Comment #2, it would be interesting to see how previously described constructs such as DS-Cav1, DS-CavEs2, and HexaPro compare to these new designs in terms of their REU, as displayed in Fig S1.

Our computational methodology was not specifically designed to enforce the formation of disulfide bonds, which are crucial components within the DS-Cav1 and DS-CavEs2 immunogens. Achieving accurate modeling of disulfide bonds requires extensive sampling of the backbone structure to meet the geometric constraints of the bond. Consequently, it becomes challenging to discern whether potential variations in energy are attributable to the disulfide bond itself, the inability to achieve optimal geometric configurations, or the potential introduction of artifacts through extensive structural sampling. In an independent study, we are establishing a disulfide protocol for these purposes.

On the other hand, the total energy of the HexaPro construct in the postfusion state does not accurately reflect the impact of the four prolines added to S-2P. This limitation arises because only one position, namely residue 942, among the four substituted residues (F817P, A892P, A899P, A942P) is found on a defined region of the crystal postfusion structure (PDB ID: 6xra). Hence, considering that just one out of four mutations can be modeled in postfusion, the comparison between the total energy of the prefusion and postfusion states, as depicted in Supplementary Figure 1, would be severely unbalanced.

Furthermore, like the challenges faced in modeling disulfide bonds, it is difficult to accurately estimate energy changes caused by proline residues because of the geometric constraints inherent to this specific amino acid. Nevertheless, we modeled the HexaPro construct in the prefusion state and compared its energy in Rosetta Energy Units (REU) to our designed proteins and the base construct S-2P. As detailed in the table below and consistent with our prior explanation, our HexaPro modeling anticipated a slight increase in the energy of the S-2P protein. A closer examination of the individual energy terms within the Rosetta scoring function revealed that this increased energy primarily resulted from a high score in the backbone torsion preference of residues preceding proline (ram_prepro term). Therefore, the observed energetic changes result from not meeting the geometric requirements around the proline mutations rather than the interactions among amino acids. Note, below values are the total energy, in the main manuscript, we work with normalized values which depend on the energy of postfusion state.

Protein variant	Total Score (REU)	Rama_prepro (REU)
HexaPro	-11985	152.81
Spk-M	-12028.8	89.289
Spk-F	-12044.7	85.766

Spk-R	-12067.2	84.816
S-2P	-11999.5	117.167

5. Because the novel MPV F and SARS-CoV-2 S stabilizing mutations reported in this manuscript were layered on top of base constructs which had already undergone some degree of stabilization, some discussion of whether these new mutations would be sufficient to stabilize the prefusion conformations of MPV F and SARS-CoV-2 S in the absence of the 115-BV and S-2P mutations would be a welcome addition.

The pre-existing mutations in the hMPV F and SARS-CoV-2 S proteins were essential for obtaining the prefusion state structure, despite these proteins being still unstable and difficult to produce. The base constructs primarily have proline residues situated at the N-terminal end of helices crucial for the postfusion transition. Due to proline's unique geometric constraints, it becomes challenging to estimate the energy changes that might result from reverting to original residues, as more extensive backbone modeling is necessary. This increases the potential for prediction errors and can introduce biases if we use the modeled structure to evaluate our protocol. Nevertheless, given the mutations' placement, we speculate that our protocol would likely have recognized these positions as a target for optimization due to the absence of helix capping and its high potential in disruption of the postfusion structure. We have included this discussion on page 8, lines 3-17.

Reviewer #2 (Remarks to the Author)

Summary

In the study the authors have developed a computational method for designing stable viral pre-fusion constructs for a range of viruses and then validated in vitro and in vivo.

Major comments

1. More could potentially be done in the mouse model. It would be interesting to look at the impact on challenge with RSV infection – disease and peak viral load. It would also be interesting to compare against a less stable/ post fusion recombinant F protein to demonstrate that the approach is an improvement. Potentially compare to sera after infection too.

While the mouse model is widely used for assessing the antibody response following vaccination, it is important to note that this animal model is only partially permissive for human RSV infections. Consequently, when the animals are subjected to experimental viral challenge, they typically do not manifest any disease symptoms or pulmonary pathology (Taylor, Vaccine, 2017, doi: 10.1016/j.vaccine.2016.11.054). Thus, a challenge study in a mouse model tends to not provide adequate insights into the effectiveness of the vaccination as mice do not really get sick by RSV.

The primary goal of our vaccination study was to assess whether the mutations introduced in the R-1b design maintained the immunological characteristics of the prefusion state. To achieve this, our focus was comparing the immune response induced by our design with that elicited by the prefusion-stabilized DS-Cav1 protein. DS-Cav1 has an established reputation of being a superior immunogen compared to the postfusion protein (McLellan et. al., Science, 2013, doi: 10.1126/science.1243283),

and we did not observe significant differences in the immune responses between DS-Cav1 and R-1b. While conducting comparative vaccination studies with the postfusion state remains relevant to further support the potential of R-1b, the current results effectively address the main objective of our study.

Minor comments

1. Page 1 line 29-31 in abstract needs re-wording 'allows identification and selection'.

This correction was included on Page 2, line 15, as "allows the identification and modification."

2. Page 12 line 5 says 10⁵ cells/well. Presumably is supposed to be 10⁵

This correction was included on Page 16, line 12.

3. Figure 4 and Mouse immunization section of methods (page 11, line 24)

Dosing schedule could do with being clearer. Presumably it is 2 μ g of the DS-Cav1 and 0.2 μ g of R-1b protein and these same doses at prime and boost. Figure legend on Fig 4 says 'using 0.2 μ g doses. Which is it?

We evaluated both 2 μ g and 0.2 μ g doses of the soluble proteins. We included Supplementary Table 7 to provide a more detailed explanation of the doses and the respective vaccination groups. In Figure 4, we presented the results of vaccinations involving the 0.2 μ g dosage, while the Supplementary Figure 11 showed the higher dose results (2 μ g). As illustrated in Figure 4, neutralization assays were only tested using serum samples from mice vaccinated with the 0.2 μ g doses. This decision was based on the observation that immunization with R-1b and DS-Cav1 at this dosage yielded no significant differences in neutralization.

Reviewer #3 (Remarks to the Author)

Summary

Gonzalez et. al. describe a general computational strategy for stabilizing class I fusion proteins by avoiding explicit negative design but using the undesired conformation as a guide to identify suboptimal positions and additional sequence optimization for the conformation of interest. As the authors discuss in their conclusion, the approach needs both a prefusion and a postfusion structures, the approach is quick and generally applicable and brings an elegant solution to stabilizing class I fusion proteins.

Major comments:

1. A shortcoming of the paper is the lack of detail on the analysis:

For instance, in the first example that they describe (RSV F), the authors state that about 40 - 50 positions displayed higher stability in the prefusion state than in the postfusion state. It would be helpful if these data were shown in a Figure or supplementary figure or table.

This data has been incorporated into accompanying Excel spreadsheets due to the substantial number of residues for each protein. The files are labeled as "RSV", "hMPV", and "Spike_Energy_Evaluation."

The "DDG_alanine_scanning" tab presents the energy changes resulting from alanine substitutions. The 40-50 positions that displayed higher stability in the prefusion state than in the postfusion state have been highlighted in green. The criteria used to identify these positions are outlined alongside the results. Referral to these Excel spreadsheets was included on page 4, lines 15-17.

Only the 7 (best?) mutations are shown. The authors should try throughout the manuscript to show the complete analysis. Would it be possible to show a clear annotated larger part (or complete) heatmap? The data will allow the appreciation of the selection of the mutations and also allow comparison with known stabilizing point mutations that are previously described at e.g. positions 155, 190, 207, 215, 486 and 487.

Within the Excel spreadsheets named "RSV", "hMPV", and "Spike_Energy_Evaluation," specifically under the tab titled "Heatmap_all_aa_scanning," were incorporated annotated heatmaps displaying all amino acid substitutions at the target sites identified through alanine scanning. For hMPV and SARS-CoV-2 examples, two separate heatmaps, designated as (a) and (b), were included. This distinction arises from the fact that these viruses were analyzed using two strategies: strategy (a) based on alanine-scanning energetic differences and strategy (b) based on protein dynamics.

In the "Combinatorial_design" tab, the sequences resulting from the initial iteration of the combinatorial design are shown. These sequences were characterized by a substantial number of mutations, with around 40-50 substitutions for the approach based on energetic differences and approximately 100-110 substitutions for the approach based on protein dynamics. To reduce the number of mutations, we applied a filtration process based on per-residue energetic differences compared to the native residues, in both prefusion and postfusion states. This filtering process is explained in the "Averaged_per_residue_energy" tab and allowed the exclusion of nearly half of the designable positions initially considered. The remaining positions were then reintroduced into a second round of combinatorial design.

Mutations resulting from the second design phase were subjected to further filtration, as outlined in the "Second_round_design" tab. It is important to note that, at this stage, the automated selection of beneficial mutations becomes particularly challenging. This is because synergistic mutations tend to display higher energetic changes than isolated mutations. Consequently, to avoid the inadvertent exclusion of single potentially stabilizing substitutions, we reviewed all redesigned positions and evaluated their suitability based on the criteria below:

- Mutations presented in groups, usually located in buried areas, were accepted if the group exhibited compact packing, with increments in van der Waals contacts compared to the wild-type structure. Furthermore, we ensured that the surrounding vicinity (within 10 Angstroms) remained undisturbed regarding packing and hydrogen bonds.
- In the case of the RSV and hMPV examples, we excluded hydrophobic substitutions, whether single or in groups, at the protein interface. However, for the SARS-CoV-2 spike, given the helical bundle arrangement in the inner interface, such mutations were considered acceptable only if they contributed to enhanced packing and did not disrupt adjacent hydrogen bond networks.

- For polar substitutions, we prioritized those with hydrogen bonds, verified by Rosetta's hydrogen bond energy term. Our focus was mainly on interactions at the protein interface, potential helix capping effects, and maintaining a balance of charges.
- Substitutions with the same charge as the original (e.g., replacing Arginine with Lysine) were rejected as isolated mutations. However, if the mutation was within a group, they were considered based on the enhancement of hydrogen bonds.
- The propensity of the introduced amino acid within a specific secondary structure was also used to reject potentially unfavorable mutations.

After carefully evaluating the designed mutations, we selected six to nine positions with significant interactions. This specific number of mutations was deliberate, as accurately predicting the energy changes between the prefusion and postfusion states becomes challenging with too few mutations. In such cases, the effects of beneficial interactions on the overall energy can be overshadowed by enhancements in rotamer conformations.

The mutations selected were primarily based on the ranking positions of the sequences that contained them. However, since the amino acid identities at these positions were often highly conserved, we also assessed the level of similarity among the different design options. Our goal was to ensure that the designs selected for testing exhibited a degree of diversity.

The above information was included on page 4, third and fourth paragraphs, and in the Material and Methods section "Selection of top designs".

2. The advantage of the approach is that very few variants were made and investigated, but a drawback is that the amount of mutations are relatively high and individual contribution of each mutation still needs to be studied which will increase the amount of variants to be generated:

An immunogen preferentially contains a minimum amount of mutations. This problem should be addressed. Therefore, a systematic analysis of the contribution of each of the potentially stabilizing mutations still needs to be performed. Moreover, in the current study, large amount of potentially stabilizing mutations (7) are described but no individual comparison with other known stabilizing point mutations are shown. If a minimal amount of mutations need to be selected, the authors should rank the effect on expression and quality and compare that with known stabilizing mutations.

The primary objective of our study was to efficiently engineer stabilized protein variants through a computational protocol, minimizing the number of variants that needed experimental testing. We agree that in the context of immunogen design, the fewer mutations, the better. In our work, we conducted extensive *in silico* evaluations and employed a combinatorial approach that allowed us to consider synergistic effects between mutations. This approach is essential since these synergistic effects may not be readily apparent through individual mutation assessments.

While we agree that computational methods are not without some margin of error in estimating energetic contributions, we illustrated that these methods are good enough to combine a few mutations and create substantially improved immunogens. Importantly, these mutations were strategically focused on positions predominantly buried within the protein structure, with minimal anticipated impact on the antibody response.

While we acknowledge that an in-depth analysis of individual mutations would provide valuable insights into the protein, we believe such an endeavor falls outside the scope of this particular study.

3. For their first example, RSV F, an energetic gap of at least 119 was selected. For RSV the variant of 119.1 REU was superior to the variants with higher REU. This means it is not straight forward to select an energetic gap and perhaps variants with lower gaps might have superior properties. Also for HMPV F, it was the variant with a lower REU gap that was superior over variants with highest REU gap.

Could interesting designs have been missed based on the arbitrary energetic gap cut off? It would be helpful if a longer list of solutions would be listed in a supplementary table with lower energetic gaps. Perhaps there are solutions with fewer mutations that still result in a well behaving protein.

We agree that it is indeed feasible to discover proteins that are equally well stabilized or exhibit even more favorable behavior. The mutational data indicates there are several combinations possible, as does the literature. For those interested in exploring further variants, we have included all the energetic mutational data we obtained in the supplementary Excel spreadsheets "RSV", "hMPV", and "Spike_Energy_Evaluation". The sequences obtained after the second round of combinatorial design can be found on the "Second_round_design" tab. As we performed a rigorous selection process for the final mutations, we have provided the complete set of mutations discovered rather than limiting the data solely to our chosen selections.

4. For RSV F R-1b was selected but no data is shown why this protein was superior over the others and how much better it was compared with the other RSV F candidates. Please add such data that justifies the selection. Apart from that, only four designs are shown.

The other RSV F candidates aggregated in solution, and their SEC profiles showed negligible signals spread across the elution. As a result, we opted not to incorporate these plots into Figure 2 to maintain visual clarity. Nonetheless, we have explained this omission on page 5, lines 8-9.

We have presented three to four designs as this represents the extent of the designs tested in our study. This limited number is because, within these selected candidates, we successfully identified stable prefusion proteins. Consequently, we considered that our proof-of-concept validation met our intended objective.

5. For HMPV, there are 7 differences between the best and second best. More analysis are needed to tease out the bare minimum of stabilizing mutations and to identify the mutations that really matter.

Our computational approach, which assesses the synergy among groups of mutations, might experience reduced accuracy when the number of mutations is low. In such cases, it becomes challenging to distinguish the energetic changes resulting from individual substitutions from those attributed to improved rotamer conformations. For this reason, we maintained a relatively high number of changes.

Additionally, it is important to note that hMPV designs were obtained with two different approaches: (a) based on energetic differences alone or (b) incorporating the protein dynamics. While both methods may identify some common mutations, our goal was to ensure that the designs selected for testing exhibited some degree of diversity.

6. Page 3, line 41: HMPV F is compared to 115-BV. Perhaps a better comparison would have been the variant described in the Hsieh paper (Nat Comm 2022).

We included the comparison with the DS-CavEs and DS-CavEs2 proteins regarding expression and thermal stability on page 5, lines 15-18, and paragraph 4 (Fig 2.D, E, F and Supplementary Fig 3 C-E). Our results revealed that expression levels were equivalent between M-104 and DS-CavEs2. However, Hsieh's designed constructs exhibited greater resistance to heat. This result is expected due to the presence of two and four non-native disulfide bonds in DS-CavEs and DS-CavEs2, respectively.

To ensure a more equitable assessment of thermal stability, we also extended our evaluation to include a comparison between our spike variant, Spk-M, and the HexaPro construct, as described by Hsieh et al. in *Science* (2020). Both proteins exhibited equally robust preservation of the prefusion conformation at elevated temperatures and demonstrated similar expression levels, as depicted in Fig 2. G, H, I and Supplementary Fig 3 A-B. This comparison was included on page 5, lines 12-15, and paragraph 3.

7. Page 3, last sentence: the current vaccines are not based on soluble S2P ectodomains but full-length variants. There is no evidence that Spk-M and F are superior to full-length proteins and whether they are better immunogens. Therefore, be more precise and delete the words 'current SARS-CoV-2 vaccine'

This mistake was corrected on page 5, line 28. We described the S-2P construct as the parent construct of our designs.

8. Page 4 , line 36: The structural variation in the apex are large. Since the apex is the most important target for neutralizing antibodies, more evaluation is needed to evaluate the impact. A detailed affinity measurement of D25 and other site 0 binders to R-1b and DS-Cav1 is important to evaluate the best design to stabilize this region in the most native state able to bind and induce apex antibodies.

The variations observed at the apex of our design come from the comparison with a crystallized protein bound to the neutralizing antibody 5C4 (PDB ID 5w23). This antibody interacts with site Ø and induces conformational changes in the region based on its binding angle approach (Tian et al., *Nat Comm*, 2017, doi: 10.1038/s41467-017-01858-w). However, without a binding partner, the antigenic site Ø has been shown to exist as an ensemble of conformations, as described by Jones et al. in *PLoS Pathog*, 2019 ([10.1371/journal.ppat.1007944](https://doi.org/10.1371/journal.ppat.1007944)). This inherent flexibility allows the binding of various antibodies utilizing distinct binding modes. This explanation was included on page 6, lines 19-28.

To provide further context, we have included Supplementary Figure 6, which compares the apex region of our design with two different crystal structures of the DS-Cav1 protein (PDBs 4mmu and 5k6c, as described by McLellan et al. in *Science*, 2013, and Joyce et al. in *Nat Struct Mol Biol*, 2016, respectively). As depicted in Supplementary Figure 6, the apex region of DS-Cav1 aligns perfectly with R-1b in its natural, unbound state. Furthermore, our ELISA analysis of prefusion-specific antibodies after vaccination with DS-Cav1 and R-1b demonstrated that these immunogens induce comparable levels of prefusion antibodies (Fig 4. B-C and Supplementary Fig 11). Additionally, the neutralization capabilities of both antibody responses were found to be equivalent (Fig 4.D). Based on these results, we believe that the conformation of the antigenic site Ø in R-1b represents a natural state, and it is unlikely to affect the immunogenic properties of the protein.

9. The approach predicts mutations that destabilize the postfusion state. However, unlike e.g. VSV-G, the conformational change of the described proteins is unidirectional and none of these proteins are known to change back from postfusion to prefusion conformation. Therefore, although the calculations based on ddG of prefusion and postfusion states is relevant and delivers successful stabilizing mutations, it is not because the postfusion structure is destabilized but because the prefusion structure is stabilized. Residues that the authors claim are postfusion disrupting are making favorable interactions in the prefusion conformation according to the structure data that the authors describe in their manuscript.

That is correct. The primary mechanism by which the selected mutations stabilize the prefusion conformation is through interactions within the prefusion state. In this context, the postfusion state serves as a reference to guide the optimization process. Compared to the postfusion state, the residues identified are not inherently optimized for the prefusion conformation. We have revised our claim that the mutations (N175R in R-1b, L130D in M-104, and T941D in Spk-M) were intended to disrupt the postfusion conformation, but they also stabilize the prefusion state. We did however measure binding to the postfusion antibody 131-2A after heating the protein (which typically results in the conformational change) and binding was substantially decreased, indicating destabilization of the postfusion state. We rephrased our wording to clarify this. Page 6/7, lines 47 – 7.

10. Residues that are surface exposed in the prefusion conformation can apparently have a stabilizing effect on the prefusion conformation (although not preferred for vaccine components.) Authors should indicate which of the mutations are surface exposed and for immunogen design it would be helpful to show which of the exposed mutations can be omitted from the current designs.

This information was included on page 8, second paragraph.

Minor comments:

1. Page 2, line 21-25: wrong reference to Cell Reports papers by Rutten et. al. The first reference (19) is likely Rutten et. al., Cell Reports 23, 584-595, 2018, in which mutations of buried charges is described for the first time.

Reference (19) was corrected.

2. A successful analysis described in the paper depends on the resolution of the structures that are used. Why was 5w23 used for the analysis for RSV prefusion F. Is it the structure with highest resolution?

We aimed to implement the computational approach in crystal structures that closely retained the native sequence. In the case of RSV F, we chose the PDB 5w23 due to the absence of engineered mutations. This protein was trapped in the prefusion state through co-crystallization with a neutralizing antibody.

Reviewer #4 (Remarks to the Author):

Summary

Thank you for the opportunity to read and review the manuscript "A general computational design

strategy for stabilizing viral class I fusion proteins." The authors have developed a Rosetta-based computational approach that seeks to stabilize the prefusion conformation by learning about suboptimal contacts relative to its alternate postfusion conformation. The algorithm allows the identification of these regions, and their potential substitutions, based on energy differences and relative motion between the two states. The computationally designed sequences are then validated by studying the immunogenicity of one design in a mouse model and in vitro neutralization and specific serum IgG patterns compared to a clinical candidate for two other cases.

I am writing this review from the point of view of someone with expertise in computational protein design, engineering, and software development. I do not have the expertise regarding the experimental procedure included in the manuscript.

Thank you for sharing the code, which will ensure reproducibility and transparency. While I have thoroughly enjoyed reading the well-written manuscript, I have the following few comments and queries:

Comments

1. Page 8, line 27: Finally, the combinatorial sequence design was carried out by the FastDesign. Is the FastDesign with a fixed backbone, or flexibility in the backbone is allowed here?

FastDesign was carried out allowing sampling of backbone torsions. This information is now more explicitly included on page 11, line 23.

2. Fig 2I: What do the two peaks of SPK-R in DSF plots signify?

The presence of multiple peaks in DSF plots can manifest in various scenarios. Some examples are the unfolding of different domains at varying temperatures, diverse oligomeric states, or conformational changes yielding distinct thermal stabilities. In the case of the spike protein, it is plausible that any of these situations contributed to the appearance of the two peaks.

The prefusion state of the spike protein exists as a trimer formed by two subunits. Within this trimer, there is structural heterogeneity, with the receptor binding domain (RBD) oscillating between "up" and "down" positions. Consequently, the application of heat during DSF experiments may have prompted a fraction of the Spk-R proteins to assume less stable conformations overall or conformations allowing partial unfolding within specific domains (such as the RBD). The heat could have also led to the dissociation of individual subunits or protomers within the trimeric structure. It reflects a change in the fold in some form, though it is not high enough in resolution to tell.

3. Due to proximity to the cellular membrane, will there be any benefit of using membrane-informed energy functions or terms instead of energy functions for soluble proteins?

We are only using the soluble part, and that is the part that has been crystallized. As the soluble forms of these proteins have proven effective in eliciting robust immune responses, our computational approach aimed to enhance the structure of these soluble variants. Since these protein variants are not expected to interact with cellular membranes, modeling membrane interactions is not essential in our current design objectives. However, it is noteworthy that in the context of vaccine delivery platforms where the full-length protein is required, including its transmembrane domain, incorporating

membrane-informed energy functions can be valuable in ensuring the accurate representation and positioning of the protein within a membrane environment.

4. A few comments on the challenges and benefits of this strategy of protein vaccine design relative to other strategies might be helpful.

There are several papers where rosetta-based energy and structural measures are used to design new proteins. It is not very clear about the novelty of this approach. Mentioning the challenges faced by current methods and the improvements attained by this approach is needed.

Class I fusion proteins have historically been stabilized through manual structure-based design, which often requires the testing of hundreds of mutations. While researchers may use computational tools to model specific mutations, there are no reports of a complete systematic approach that guides the design process from the beginning to the end. However, as the multi-state design idea inspired our approach, we included the main benefit of our implementation on page 7, lines 31-40, compared to other multi-state design strategies. The challenges faced by our strategy are described on page 7, last paragraph.

The main challenge with rosetta-base protocols involving multiple protein stages is the elevated computational cost of the sequence optimization while modeling all states simultaneously. This condition often limits the sampled sequence space and renders sub-optimal solutions. With our design strategy, we reduce the complexity of the sequence search by focusing only on modeling and optimizing one structure, while the undesired conformation defines the potential mutations to be sampled. We included this comment on page 7, lines 31-40.

REVIEWERS' COMMENTS

Reviewer #1 (Remarks to the Author):

Many thanks to the authors for their thorough response to the initial review.

Reviewer #2 (Remarks to the Author):

Thanks for making the revisions suggested

Mice can show signs of disease with RSV, but outside of the scope of this project, so doesnt need addressing